# Microtubules originate asymmetrically at the somatic golgi and are guided via Kinesin2 to maintain polarity within neurons

Amrita Mukherjee[1], Paul S Brooks[1], Fred Bernard[2], Antoine Guichet[2], Paul T Conduit[1,2]*

[1]Department of Zoology, University of Cambridge, Cambridge, United Kingdom; [2]Université de Paris, CNRS, Institut Jacques Monod, Paris, France

**Abstract** Neurons contain polarised microtubule arrays essential for neuronal function. How microtubule nucleation and polarity are regulated within neurons remains unclear. We show that γ-tubulin localises asymmetrically to the somatic Golgi within *Drosophila* neurons. Microtubules originate from the Golgi with an initial growth preference towards the axon. Their growing plus ends also turn towards and into the axon, adding to the plus-end-out microtubule pool. Any plus ends that reach a dendrite, however, do not readily enter, maintaining minus-end-out polarity. Both turning towards the axon and exclusion from dendrites depend on Kinesin-2, a plus-end-associated motor that guides growing plus ends along adjacent microtubules. We propose that Kinesin-2 engages with a polarised microtubule network within the soma to guide growing microtubules towards the axon; while at dendrite entry sites engagement with microtubules of opposite polarity generates a backward stalling force that prevents entry into dendrites and thus maintains minus-end-out polarity within proximal dendrites.

**\*For correspondence:**
paul.conduit@ijm.fr

**Competing interests:** The authors declare that no competing interests exist.

## Introduction

Microtubules are polarised α/β-tubulin-based polymers essential for cell viability, providing pushing and pulling forces, structural support or tracks for the transport of intracellular cargo (*Goodson and Jonasson, 2018*). α-tubulin is located at the so-called minus end and β-tubulin is exposed at the so-called plus end, which is typically more dynamic. This inherent microtubule polarity is important for cell polarity, as different motor proteins (Kinesins and Dynein) move cargo along microtubules in a specific direction – Dynein towards the minus end and most Kinesins towards the plus end. In neurons, most plus ends point away from the soma in axons (plus-end-out microtubules), while the microtubules in dendrites are either of mixed polarity or are predominantly minus-end-out (*Hill et al., 2012*; *Kapitein and Hoogenraad, 2015*; *Kelliher et al., 2019*; *Tas et al., 2017*). This difference between axons and dendrites is important for the correct distribution of cargo throughout the neuron (*Harterink et al., 2018*; *Kapitein and Hoogenraad, 2015*; *Tas et al., 2017*).

Within cells de novo assembly of new microtubules, that is microtubule nucleation, is kinetically unfavourable and is templated and catalysed by multi-protein γ-tubulin ring complexes (γ-TuRCs) (*Tovey and Conduit, 2018*). Knockdown of γ-TuRCs within model systems affects dynamic microtubules in all neuronal compartments (*Nguyen et al., 2014*; *Ori-McKenney et al., 2012*; *Sánchez-Huertas et al., 2016*; *Yamada and Hayashi, 2019*; *Yau et al., 2014*) and mutations in γ-TuRC genes have been linked to human neurodevelopmental disorders (*Bahi-Buisson et al., 2014*; *Mitani et al., 2019*; *Poirier et al., 2013*). γ-TuRCs are typically inactive until they are recruited to specific sites within cells, such as microtubule organising centres (MTOCs), the cytosol around mitotic chromatin,

or the sides of pre-existing microtubules via binding to Augmin/HAUS complexes (*Farache et al., 2018*; *Lin et al., 2015*; *Meunier and Vernos, 2016*; *Sanchez and Feldman, 2017*; *Teixido-Travesa et al., 2012*). A range of MTOCs exist, including centrosomes, the Golgi apparatus and the nuclear envelope, and different cells use different MTOCs to help generate and organise their specific microtubule arrays (*Sanchez and Feldman, 2017*). γ-TuRC recruitment occurs via γ-TuRC 'tethering proteins', such as *Drosophila* Centrosomin (Cnn), that simultaneously bind to the γ-TuRC and a particular MTOC, and can also help activate the γ-TuRC (*Tovey and Conduit, 2018*).

Although γ-tubulin is important within neurons (*Nguyen et al., 2014*; *Ori-McKenney et al., 2012*; *Sánchez-Huertas et al., 2016*; *Yamada and Hayashi, 2019*; *Yau et al., 2014*), it remains unclear how microtubule nucleation is regulated. During early development of mammalian neurons, the centrosome within the soma nucleates microtubules (*Stiess et al., 2010*) that are severed and transported into neurites via motor-based microtubule sliding (*Baas et al., 2005*). Microtubule sliding is also important for axon outgrowth in *Drosophila* cultured neurons (*Del Castillo et al., 2015*; *Lu et al., 2013*), and for establishing microtubule polarity (*del Castillo et al., 2015*; *Klinman et al., 2017*; *Rao et al., 2017*; *Yan et al., 2013*; *Zheng et al., 2008*). Centrosomes are inactivated, however, at later developmental stages (*Stiess et al., 2010*) and are dispensable for neuronal development in both mammalian and fly neurons (*Nguyen et al., 2011*; *Stiess et al., 2010*). No other active MTOCs within the neuronal soma have been described. Nevertheless, microtubules continue to grow within the soma (*Nguyen et al., 2011*; *Sánchez-Huertas et al., 2016*), and in mammalian neurons this depends in part on the HAUS complex (*Sánchez-Huertas et al., 2016*), which is also important for microtubule growth within axons and dendrites (*Cunha-Ferreira et al., 2018*; *Sánchez-Huertas et al., 2016*). Some MTOCs have been identified within dendrites: the basal body, or its surrounding region, within the distal part of *C. elegans* ciliated neurons acts as an MTOC, as does a similar region within the URX non-ciliated neuron (*Harterink et al., 2018*); an MTOC made from endosomes that tracks the dendritic growth cone in *C. elegans* PVD neurons has recently been identified (*Liang et al., 2020*); and fragments of Golgi called Golgi outposts within the dendrites of *Drosophila* dendritic arborisation (da) neurons are thought to recruit γ-TuRCs and act as MTOCs (*Ori-McKenney et al., 2012*; *Yalgin et al., 2015*; *Zhou et al., 2014*).

*Drosophila* larval da neurons are a popular *in vivo* neuronal model (*Jan and Jan, 2010*). Ease of imaging and genetic manipulation coupled with the ability to examine different neuronal classes, each with a stereotypical dendritic branching pattern, make them a model of choice when examining microtubule organisation within neurons. Four classes exist, including class I neurons that are proprioceptive and have the simplest 'comb-like' dendritic branching pattern, and class IV neurons that are nociceptive and have the most elaborate branching pattern, tiling the surface of the larva (*Figure 1—figure supplement 1*; *Grueber et al., 2002*). In both neuronal classes, microtubules in axons are predominantly plus-end-out throughout development, but microtubule polarity in dendrites progressively becomes more minus-end-out as the neurons develop; at two days post larval hatching the majority of dynamic microtubules are minus-end-out (*Hill et al., 2012*; *Stone et al., 2008*). Microtubule polarity within da neurons can be disrupted when microtubule regulators are depleted (*Mattie et al., 2010*; *Nguyen et al., 2014*; *Ori-McKenney et al., 2012*; *Rolls and Jegla, 2015*; *Sears and Broihier, 2016*; *Weiner et al., 2016*; *Yalgin et al., 2015*; *Zhou et al., 2014*), but we lack a full understanding of how microtubule polarity is established and maintained.

Golgi outposts are thought to provide localised sites of microtubule nucleation within the dendrites of da neurons to help regulate dendritic outgrowth and microtubule polarity (*Ori-McKenney et al., 2012*; *Yalgin et al., 2015*; *Zhou et al., 2014*). Unlike the somatic Golgi, which comprises cis, medial, and trans compartments organised into stacks and ribbons (*Kondylis and Rabouille, 2009*; *Nakamura et al., 1995*), Golgi outposts can be either single- or multi-compartment units with multi-compartment units having a higher propensity to initiate microtubule growth (*Zhou et al., 2014*). Within class I da neurons, Golgi outpost-mediated nucleation is thought to be dependent on the γ-TuRC-tethering protein Cnn and to help restrict dendritic branching (*Yalgin et al., 2015*), while within class IV neurons Golgi outpost-mediated nucleation is believed to be dependent on Pericentrin-like protein (Plp) and to help promote dendritic branching (*Ori-McKenney et al., 2012*). There remains some uncertainty, however, about the role of Golgi outposts in microtubule nucleation, as a separate study that examined the localisation of ectopically expressed γ-tubulin-GFP concluded that γ-tubulin localises predominantly to dendritic branch points in a Golgi outpost-independent manner (*Nguyen et al., 2014*).

In this study, we initially aimed to identify sites of microtubule nucleation within *Drosophila* da neurons. We used endogenously tagged γ-tubulin-GFP as a proxy for γ-TuRCs and performed a detailed analysis of γ-tubulin localisation within both class I and class IV da neurons. We find variation between neuronal classes in how γ-tubulin localises within dendrites, but find that the most prominent localisation of γ-tubulin is at the cis-compartment of somatic Golgi stacks within all sensory neurons of the dorsal cluster. This asymmetric γ-tubulin-Golgi association is not dependent on either Cnn or Plp, suggesting another molecule must regulate γ-TuRC recruitment to the Golgi. Tracking of EB1-GFP comets within the soma suggests that microtubules are nucleated asymmetrically from the somatic Golgi. We find that these Golgi-derived microtubules initially grow preferentially towards the axon. During growth, they are also guided towards the axon and away from dendrites in a Kinesin-2-dependent manner, and they readily enter the axon contributing to plus-end-out microtubule polarity. In contrast, the relatively small number of microtubules that do grow towards a dendrite are normally excluded and this also depends upon Kinesin-2. After Kinesin-2 depletion, growing microtubules more readily approach and enter the dendrites causing a dramatic increase in the proportion of anterograde microtubules in proximal dendrites. This results in a reversal of overall microtubule polarity in proximal dendrites from predominantly minus-end-out to predominantly plus-end-out. We therefore propose that plus-end-associated Kinesin-2 guides growing microtubules towards the axon and away from dendrites along a polarised microtubule network within the soma, while at dendrite entry points Kinesin-2 generates a stalling force on growing microtubules when engaging dendritic microtubules of opposite polarity.

## Results

### A detailed analysis of endogenous γ-tubulin localisation within class I and class IV da neurons

We began by examining the localisation of γ-tubulin (as a proxy for γ-TuRCs) within class I and class IV da neurons. To avoid any potential artefacts induced by ectopic overexpression, we used alleles where γ-tubulin23C (the zygotic form of γ-tubulin) was tagged at its endogenous locus with GFP (*Tovey and Conduit, 2018* and this study). We generated fly stocks expressing two genetic copies of endogenously-tagged γ-tubulin23C-GFP (hereafter γ-tubulin-GFP) and the membrane marker mCD8-RFP, expressed either in class I or class IV da neurons, and imaged living animals. The most striking and obvious localisation of γ-tubulin-GFP was as multiple bright and relatively large puncta within the soma of both neuronal types (*Figure 1A*; *Figure 2A*); we address this localisation in subsequent sections. We could also detect discrete accumulations of γ-tubulin-GFP at specific dendritic sites (*Figure 1A*; *Figure 2A*), which were typically dim and varied between class I and class IV neurons. We therefore describe the localisation of γ-tubulin-GFP within dendrites for each neuron type in turn below.

For class I da neurons we focussed on the ddaE neuron (hereafter class I neurons), as these have been the most widely characterised. From 13 class I neurons analysed, we detected an average of 1.1 discrete γ-tubulin-GFP puncta per 100 μm of dendrite (*Figure 1A*). Most puncta were just above background intensity levels, although some were bright, and they were found more frequently away from branchpoints (80.2% of puncta) than within branchpoints (19.8% of puncta). We could detect γ-tubulin-GFP signal in 18% of branchpoints; in approximately half of these branchpoints the γ-tubulin-GFP appeared as puncta, often with multiple puncta per branchpoint, while in the other half of branchpoints the γ-tubulin-GFP signal was spread more diffusely through the branchpoint (*Figure 1B*). This diffuse signal was similar to that observed when γ-tubulin-GFP is overexpressed (*Nguyen et al., 2014*; *Weiner et al., 2020*; *Figure 1—figure supplement 2A*), although it appeared that the frequency of branchpoint occupancy and the strength of the endogenous γ-tubulin-GFP signal was lower than that observed with ectopically expressed UAS-γ-tubulin-GFP. It is possible that the localisation of endogenous γ-tubulin-GFP at branchpoints represents the recently reported recruitment of γ-tubulin to endosomes at branchpoints that provide a platform for microtubule nucleation (*Weiner et al., 2020*). We have not tested here, however, whether the diffuse or punctate γ-tubulin-GFP signals, or both, are functionally important at branchpoints.

We noticed that class I neurons display 'bubbles', where the diameter of the dendrite is locally increased to differing extents (*Figure 1A*); there were on average 2.6 'bubbles' per 100 μm of

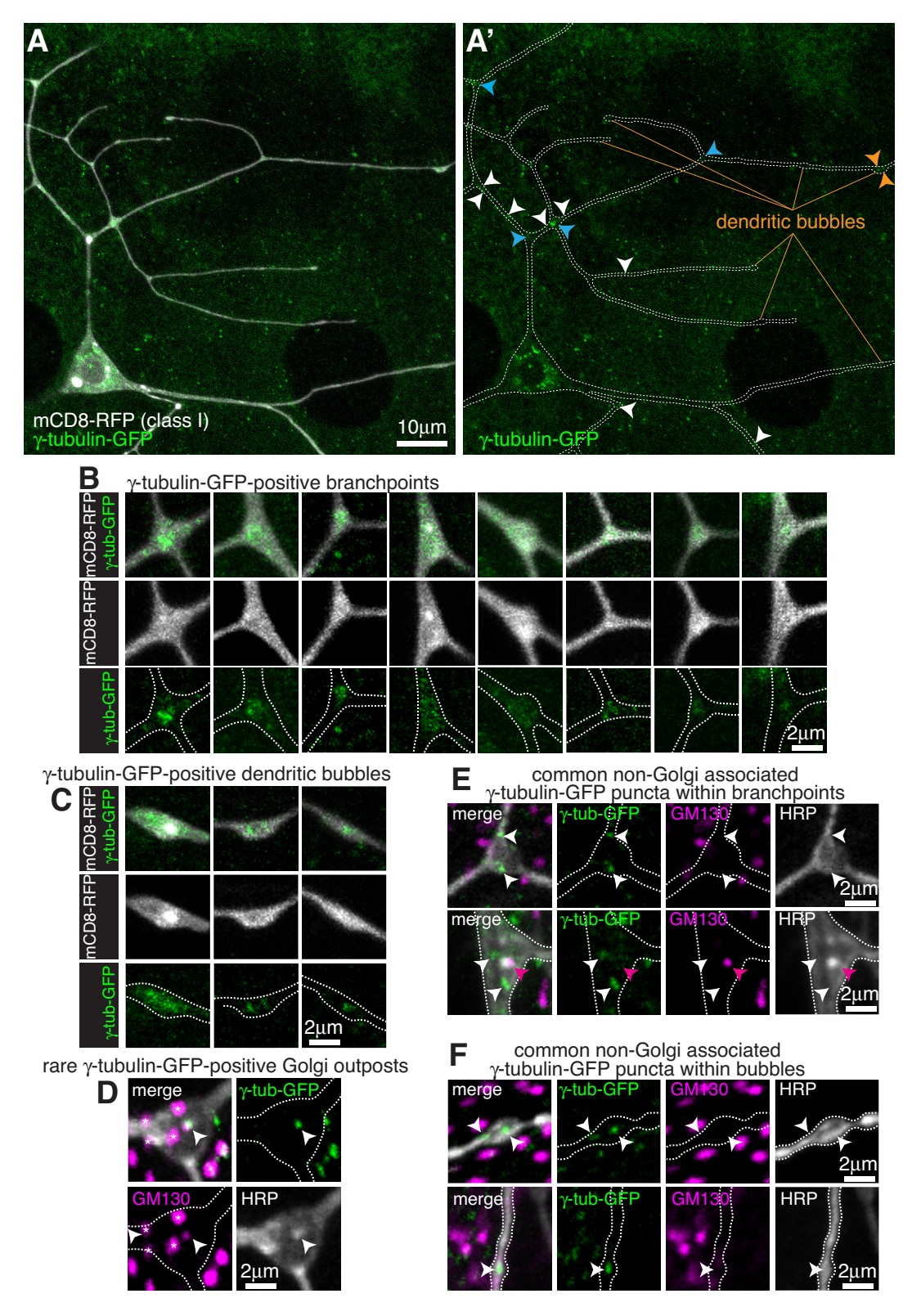

**Figure 1.** Endogenously-tagged γ-tubulin-GFP localises to a fraction of branchpoints and dendritic bubbles within class I da neurons. (**A**) Fluorescent confocal images of the proximal region of a class I da neuron expressing mCD8-RFP (greyscale) within a living 3rd instar larva expressing endogenously-tagged γ-tubulin-GFP (green). Left panel (**A**) shows an overlay of the GFP and RFP signals, right panel (**A'**) shows only the GFP signal with the outline of the neuron drawn for clarity; white, blue and orange arrowheads indicate γ-tubulin-GFP puncta/accumulations within dendritic stretches, branchpoints,

*Figure 1 continued on next page*

*Figure 1 continued*

and dendritic bubbles, respectively. (B,C) Selected images of γ-tubulin-GFP-positive branchpoints (B) or dendritic bubbles (C) from living neurons as in (A). Individual mCD8-RFP channel images (greyscale) have been included for clarity. (D–F) Confocal images show branchpoints (D,E) or dendritic bubbles (F) from 3ʳᵈ instar larvae expressing endogenous γ-tubulin-GFP fixed and immunostained for GFP (green), GM130 (magenta) and HRP (greyscale). γ-tubulin-GFP signal was rarely observed co-localising with GM130 and HRP signal (D), and frequently observed independent of the Golgi markers at both branchpoints (E) and dendritic bubbles (F).

The online version of this article includes the following figure supplement(s) for figure 1:

**Figure supplement 1.** Class I and class IV da neuron morphology.

**Figure supplement 2.** Ectopically expressed γ-tubulin-GFP is strongly enriched in branchpoints and dendritic bubbles and ectopically expressed ManII is not a reliable marker of Golgi outposts in class I da neurons.

dendrite with a larger fraction found further from the soma;~16.5% of bubbles contained γ-tubulin-GFP signal, either as weak puncta or a diffuse signal (*Figure 1C*), and ~36.7% of the γ-tubulin-GFP puncta that we observed within dendrites were found within bubbles. It remains to be tested whether the accumulation of γ-tubulin within these bubbles is functionally relevant.

To test whether γ-tubulin-GFP puncta within class I dendrites associate with Golgi outposts, we fixed and stained larval preparations expressing γ-tubulin-GFP with antibodies against HRP, which stain neuronal membranes, GFP, and the Golgi protein GM130. Out of 9 neurons from three larvae, we could only find one example where γ-tubulin-GFP colocalised with an HRP punctum, and in this case GM130 was also colocalised suggesting it was a Golgi outpost (*Figure 1D*). GM130 labels only multi-compartment Golgi outposts (*Zhou et al., 2014*) but HRP probably labels most, if not all, Golgi outposts. We chose not to use the ectopically expressed Golgi marker UAS-ManII-GFP used in some previous studies, as this has been reported to 'leak' into endosomes within the dendrites of class I da neurons (*Weiner et al., 2020*). Consistent with this, we found that 221-Gal4-driven UAS-ManII-GFP can form puncta, enrichments, and long stretches within class I dendrites that are not associated with HRP staining (*Figure 1—figure supplement 2B*). These apparent cytosolic accumulations may therefore be due to an excess of ManII protein within the dendrites. This appeared to be specific to dendrites, however, as UAS-ManII-GFP always colocalised with HRP staining within the soma (*Figure 1—figure supplement 2C*), possibly due to the higher concentration of Golgi within the soma. All dendritic γ-tubulin-GFP puncta (except for the single occasion noted above), including those within branchpoints, dendrites and bubbles, did not colocalise with HRP or GM130 staining (*Figure 1E,F*). Thus, our data strongly suggest that γ-TuRCs do not typically associate with Golgi outposts within class I neurons, and instead localise to a fraction of branchpoints and dendritic bubbles in a Golgi outpost-independent manner.

For class IV neurons we focussed on the ddaC neuron (hereafter class IV neurons), as these have also been the most widely characterised. We analysed two distinct dendritic regions that we term proximal (within ~100 µm of the soma) and distal (over ~200 µm from the soma). We detected an average of 2.3 and 0.3 γ-tubulin-GFP puncta per 100 µm of dendrite within proximal (10 neurons analysed) and distal (seven neurons analysed) regions, respectively, and the intensity of the majority was only just above background levels, including all of the puncta in the distal regions (*Figure 2A*). Nevertheless, we consistently observed bright γ-tubulin-GFP puncta within the primary and secondary branchpoints close to the soma (*Figure 2A*). 44.6% of γ-tubulin-GFP puncta observed in the proximal regions were found within the large primary and secondary branchpoints, and 57% of these branchpoints contained γ-tubulin-GFP puncta, which were always bright relative to other foci within the dendrites (*Figure 2A*). Staining with HRP and GM130 antibodies showed that these bright γ-tubulin-GFP puncta associated with Golgi outposts (*Figure 2B*). Intriguingly, γ-tubulin-GFP only colocalised with HRP puncta that also associated with GM130 (*Figure 2B*), suggesting that γ-TuRCs are recruited only to multi-compartment Golgi outposts. The high frequency of proximal branchpoints that contained γ-tubulin-GFP puncta contrasted with the near absence of γ-tubulin-GFP puncta within the smaller branchpoints of the distal arbor, where only 1.7% of branchpoints contained detectable but very weak γ-tubulin-GFP signal (*Figure 2A* insets).

In contrast to class I neurons, we found that the vast majority of ectopically expressed (via ppk-Gal4) ManII-GFP puncta within class IV neurons colocalised with HRP and did not form large accumulations or stretches within the dendrites (*Figure 2C*), suggesting that ectopically expressed ManII-GFP can be used as a reliable marker of Golgi outposts in class IV neurons. We observed an average

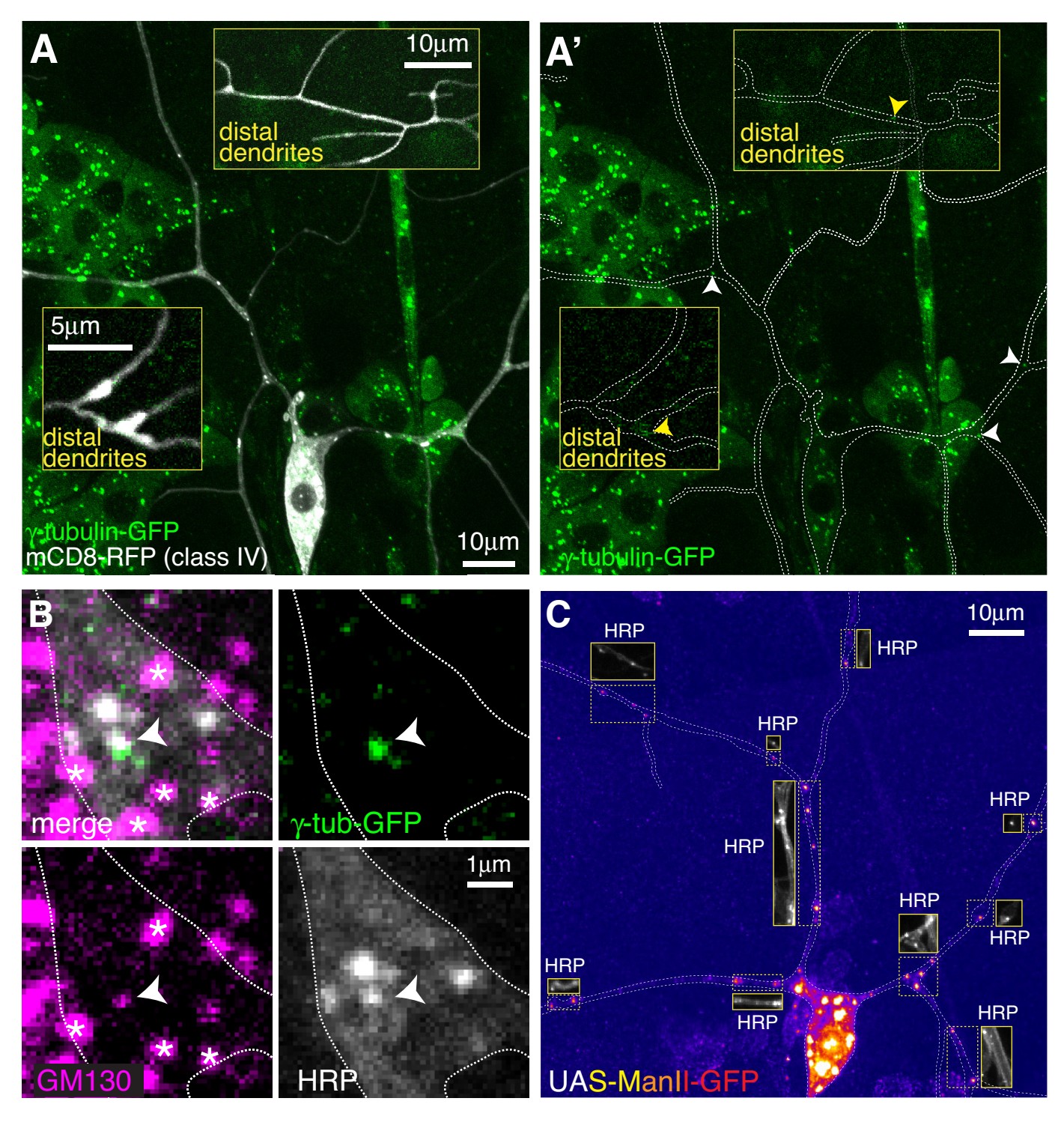

**Figure 2.** Endogenously-tagged γ-tubulin-GFP localises to a small fraction of proximal Golgi outposts within class IV da neurons. (**A**) Fluorescent confocal images of the proximal region of a class IV da neuron expressing UAS-mCD8-RFP (greyscale) within a living 3$^{rd}$ instar larva expressing endogenously-tagged γ-tubulin-GFP (green). Left panel (**A**) shows an overlay of the GFP and RFP signals, right panel (**A'**) shows only the GFP signal with the outline of the neuron drawn for clarity; insets show examples of distal dendrites located >200 μm from the soma; white and yellow arrowheads indicate bright or weak γ-tubulin-GFP puncta, respectively, within proximal or distal dendrites. (**B**) Images show a γ-tubulin-GFP-positive proximal branchpoint from a 3$^{rd}$ instar larva expressing endogenous γ-tubulin-GFP fixed and immunostained for GFP (green), GM130 (magenta) and HRP (greyscale). Asterisks indicate GM130 foci from overlapping epidermal cells. Arrowhead indicates an example of a γ-tubulin-GFP-positive Golgi outpost

*Figure 2 continued on next page*

*Figure 2 continued*

observed within proximal class IV branchpoints. (C) Confocal images show the proximal region of a class IV dendritic arbor within a 3^rd instar larva expressing ManII-GFP (to mark medial Golgi) fixed and immunostained for GFP (fire) and HRP (greyscale within insets). Insets show HRP staining alone.

of ~12 and~5 ManII-GFP-positive Golgi outposts within proximal and distal regions, respectively, which is consistent with previous observations (*Zheng et al., 2008*) and far higher than the 2.3 and 0.3 γ-tubulin-GFP puncta per 100 µm that we observed in proximal and distal dendrites (compare *Figure 2A* to *Figure 2C*). Thus, only Golgi outposts within the proximal branchpoints of class IV neurons associate readily with γ-tubulin. In contrast to class I neurons, we very rarely observed γ-tubulin-GFP spread diffusely through branchpoints, suggesting that Golgi outpost-independent accumulation of γ-tubulin at branchpoints is specific to class I neurons. Moreover, we found far fewer dendritic bubbles per 100 µm of dendrite in class IV neurons (0.4 and 0.8 per 100 µm dendrite in proximal and distal regions, respectively). Collectively, our data show that Golgi outposts within class IV neuron branchpoints close to the soma frequently associate with γ-tubulin, but that the majority of Golgi outposts do not. Whether this proximal Golgi outpost associated γ-tubulin represents fully functional γ-TuRCs remains to be tested.

## γ-tubulin-GFP localises to the somatic Golgi of sensory neurons in a Cnn- and Plp-independent manner

The most striking and obvious localisation of γ-tubulin-GFP within both class I and class IV neurons was as multiple large bright puncta within their soma (see images of neurons from live animals in *Figure 1A* and *Figure 2A*, and of fixed and stained neurons in *Figure 3A–C*). We also observed similar puncta within the somas of other nearby sensory neurons (*Figure 3A*), including external sensory (es) neurons; these sensory neurons possess basal bodies that appear to associate with large amounts of γ-tubulin-GFP (*Figure 3A*). Staining with antibodies against HRP and the Golgi marker GM130 showed that all γ-tubulin-GFP puncta within the soma of all sensory neurons associated with somatic Golgi stacks (*Figure 3B,C*; *Figure 3—figure supplement 1*), which are typically scattered throughout the cytosol in *Drosophila* cells while still maintaining cis-medial-trans polarity (*Kondylis and Rabouille, 2009*). These Golgi stacks can be oriented side-on or face-on to the imaging plane, appearing either more elongated or circular, respectively. Our staining showed that the signals of γ-tubulin-GFP and GM130 partially overlapped at side-on stacks, with γ-tubulin-GFP extending further out laterally than GM130 (*Figure 3B*; *Figure 3—figure supplement 1A*); γ-tubulin-GFP surrounded GM130 in a ring-like pattern on face-on stacks (*Figure 3C*; *Figure 3—figure supplement 1B*). To determine whether γ-tubulin associates specifically with the cis-Golgi, as has been suggested for γ-tubulin in non-neuronal mammalian cells (*Wu et al., 2016*), we stained the neurons with antibodies against HRP, GM130, and Arl1. The mammalian homologue of GM130 is a cis-Golgi protein, while Arl1 is a known trans Golgi protein in *Drosophila* (*Munro, 2011*). We found that the GM130 and Arl1 signals were offset at side-on stacks, consistent with them being cis- and trans-Golgi proteins, respectively. Moreover, γ-tubulin colocalised with GM130 rather than Arl1 (*Figure 3D*). Together, this shows that γ-tubulin-GFP, possibly in the form of γ-TuRCs, localises to the rims of the cis-Golgi stack in the soma of *Drosophila* da neurons.

Two proteins, Cnn and Plp, were possible candidates for γ-TuRC recruitment to the somatic Golgi, as their homologues are required for proper organisation of microtubules at the somatic Golgi in cycling mammalian cells (*Rios, 2014*; *Roubin et al., 2013*; *Wang et al., 2010*; *Wu et al., 2016*), and Cnn and Plp have been implicated in γ-TuRC recruitment to Golgi outposts in *Drosophila* class I neurons (*Yalgin et al., 2015*), with Plp also being implicated in class IV neurons (*Ori-McKenney et al., 2012*). Cnn is a multi-isoform gene with three sets of isoforms driven by three different promoters (*Eisman et al., 2009*; *Figure 4—figure supplement 1A*). Promoter one drives the most-studied isoform (that we term Cnn-P1) that localises to centrosomes during mitosis; promoter two drives isoforms (Cnn-P2) that are yet to be characterised; and promoter three drives isoforms that are expressed specifically within testes (*Chen et al., 2017*) and so have not been considered in this study. Immunostaining with antibodies against Cnn revealed very weak, if any, signal within the soma or dendrites of the dorsal sensory neurons; while the presumptive basal bodies of the es neurons displayed a strong Cnn signal (data not shown). Given that antibody staining can be

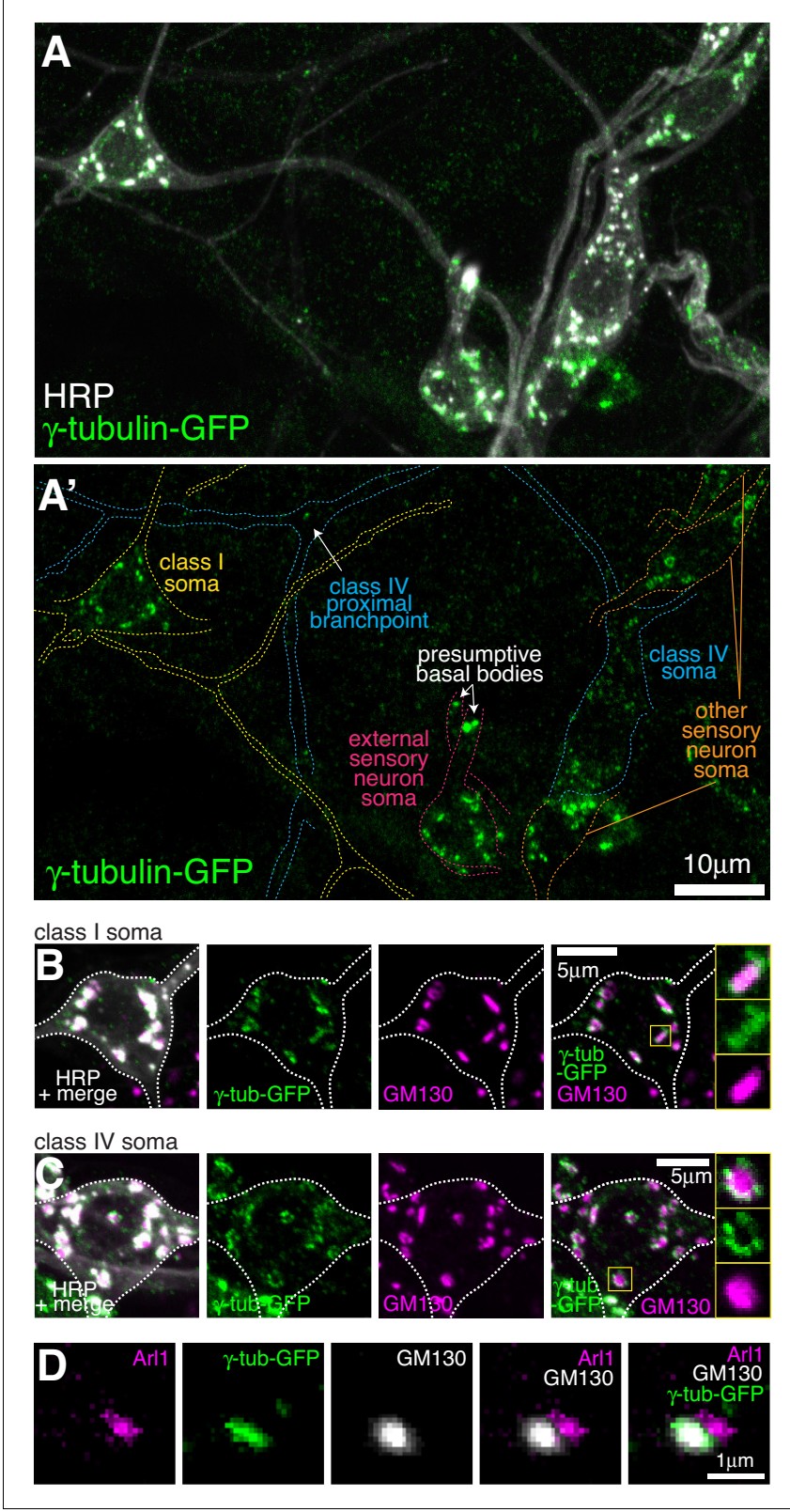

**Figure 3.** Endogenously-tagged γ-tubulin-GFP localises to the somatic Golgi of sensory neurons. (**A**) Confocal images show the somas and some proximal dendrites of sensory neurons within the dorsal cluster: class I (yellow), class IV (blue), as (pink) and other (orange) from a 3$^{rd}$ instar larva expressing endogenously-tagged γ-tubulin-GFP and immunostained for GFP (green) and HRP (marking Golgi stacks, greyscale). Upper panel (**A**) shows an overlay

*Figure 3 continued*

of the GFP and HRP signals, lower panel (**A'**) shows only the GFP signal with coloured neuronal outlines drawn for clarity. (**B, C**) Confocal images show the somas of class I (**B**) or class IV (**C**) da neurons from a 3^rd instar larva expressing endogenously-tagged γ-tubulin-GFP fixed and immunostained for GFP (green), GM130 (magenta) and HRP (greyscale). Enlarged boxes in (**B**) and (**C**) show side-on and face-on stacks, respectively. (**D**) Confocal images show an example of a single Golgi stack within a da neuron from a 3^rd instar larva expressing endogenously-tagged γ-tubulin-GFP fixed and immunostained for GFP (green), Arl1 (trans-Golgi, magenta) and GM130 (cis-Golgi, greyscale). The larva was also stained with HRP antibodies to identify neurons, but this channel has been omitted for clarity.

The online version of this article includes the following figure supplement(s) for figure 3:

**Figure supplement 1.** Endogenously-tagged γ-tubulin-GFP localises to the somatic Golgi of sensory neurons.

---

problematic, we generated flies where Cnn-P1 or Cnn-P2 were tagged with GFP at their isoform-specific N-termini (hereafter, GFP-Cnn-P1 and GFP-Cnn-P2; *Figure 4—figure supplement 1A*). The GFP insertions appear functional as flies could be kept as homozygous stocks and the localisation of GFP-Cnn-P1 to centrosomes in syncytial embryos was normal (*Figure 4—Video 1*). GFP-Cnn-P1 signal was very weak and inconsistent within the soma of living class I da neurons (*Figure 4A*). In contrast, we could readily detect clear GFP-Cnn-P1 puncta within distal dendritic bubbles in live animals (*Figure 4B*). In fixed samples, there was a weak GFP-Cnn-P1 signal associated with the somatic Golgi in the dorsal cluster of sensory neurons, similar to that observed in live samples (*Figure 4—figure supplement 1B*), although we did not detect GFP-Cnn-P1 associated with HRP puncta within dendrites (*Figure 4—figure supplement 1B*; data not shown). We found a strong GFP-Cnn-P1 signal at the presumptive basal bodies of the es neurons (*Figure 4—figure supplement 1B*). Collectively our data suggest that GFP-Cnn-P1 does not readily associate with Golgi, but accumulates within a fraction of dendritic bubbles in class I neurons, possibly together with γ-tubulin.

In contrast to GFP-Cnn-P1, we did not detect any obvious GFP-Cnn-P2 signal within the soma or dendrites of da neurons in living larvae (*Figure 4—figure supplement 1C*). Instead, GFP-Cnn-P2 appeared to be expressed within glial cells that ensheath the axons, soma, and part of the proximal dendritic region of the da neurons (*Figure 4—figure supplement 1C*). These glia are known to help regulate dendritic development (*Han et al., 2011*; *Sepp and Auld, 2003*; *Yadav et al., 2019*), and thus Cnn-P2 isoforms may have an indirect role in dendritic arbor growth. GFP-Cnn-P2 also localised strongly to the presumptive basal body of es neurons (data not shown). Consistent with the localisation pattern of GFP-Cnn-P1 and GFP-Cnn-P2, γ-tubulin-GFP could still associate with the somatic Golgi of the sensory neurons in *cnn* mutant larvae, but not to the basal bodies of es neurons (*Figure 4C*). We therefore conclude that Cnn is dispensable for γ-TuRC recruitment to the somatic Golgi within sensory neurons.

In contrast to Cnn, antibodies against Plp readily stained the somatic Golgi in all sensory neurons, including class I and class IV da neurons (*Figure 5—figure supplement 1A*). The Plp signal was, however, offset from the γ-tubulin-GFP signal at Golgi stacks in both class I and class IV da neurons (*Figure 5A,B*), suggesting that they localise to different Golgi compartments. Plp also associated with the class IV proximal Golgi outposts (*Figure 5C*) and the presumptive basal bodies of the es neurons (*Figure 5—figure supplement 1B*), and was enriched within a fraction of the distal class I dendritic bubbles (*Figure 5—figure supplement 1C*). Strikingly, γ-tubulin-GFP localisation at the somatic Golgi of all sensory neurons, including class I and class IV neurons was unaffected in *plp* mutant larvae (*Figure 5D*). This was also true of the γ-tubulin-positive proximal Golgi outposts in class IV neurons (*Figure 5D*). The absence of Plp did, however, lead to the loss of γ-tubulin-GFP from the basal bodies of es neurons (*Figure 5D*). In summary, neither Plp nor Cnn are required for the efficient recruitment of γ-tubulin to the somatic Golgi within sensory neurons, or to the few γ-tubulin-GFP-positive Golgi outposts within the proximal branch points of class IV da neurons, but both are required for the localisation of γ-tubulin-GFP to the basal body region within es neurons.

## Growing microtubules originate asymmetrically from the somatic golgi

We next wanted to assess whether the somatic Golgi is an active MTOC. EB1-GFP binds to growing microtubule ends and generates fluorescent 'comets', typically representing growing microtubule plus ends. While the origin of these comets can represent either microtubule regrowth after

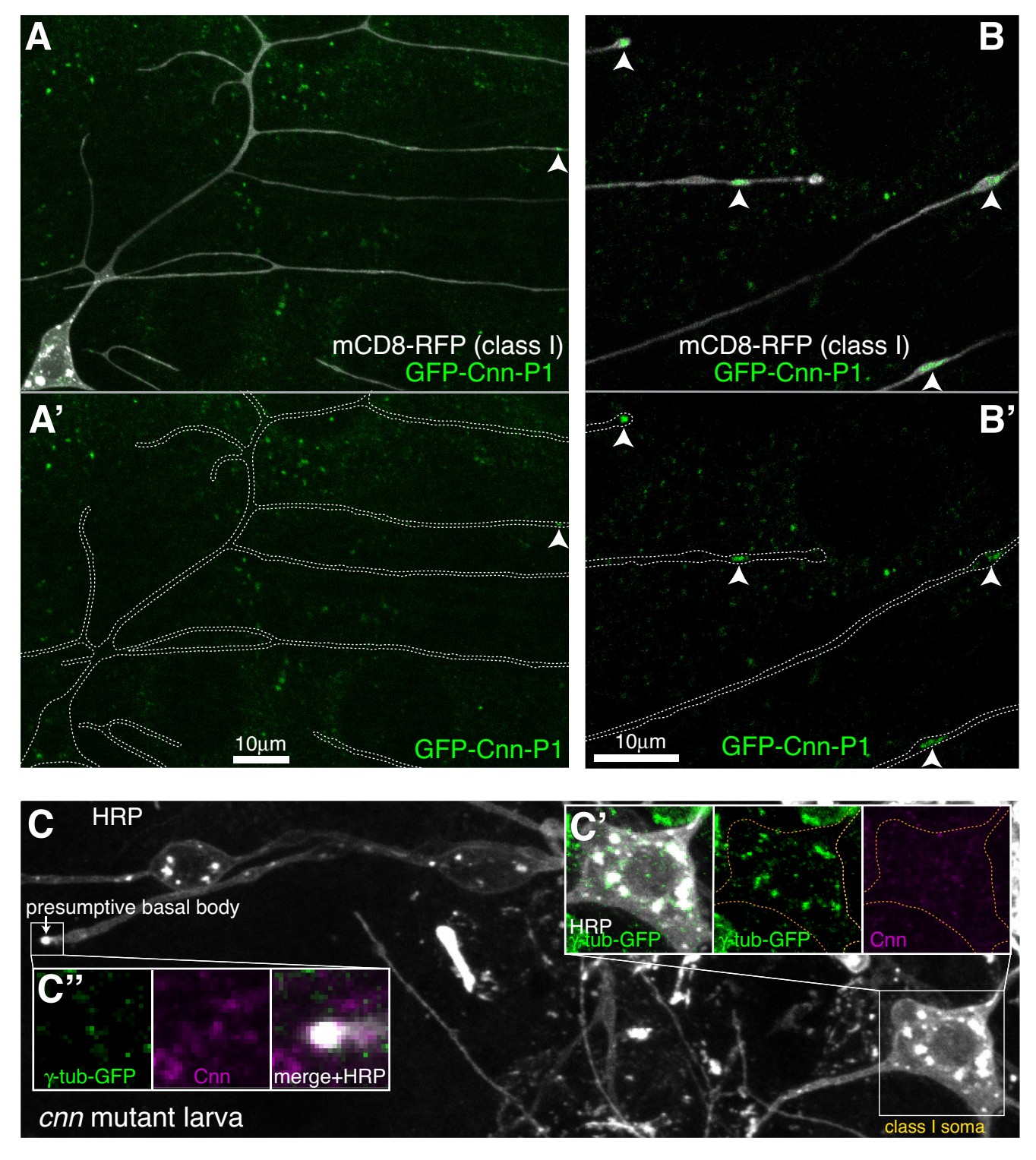

**Figure 4.** Cnn-P1 is dispensable for γ-tubulin-GFP recruitment to the somatic Golgi. (A,B) Fluorescent confocal images of proximal (A) and distal (B) regions of class I da neurons expressing mCD8-RFP (greyscale) within a living 3rd instar larva expressing endogenously-tagged GFP-Cnn-P1 (green). Upper panels show an overlay of the GFP and RFP channels, lower panels show only the GFP channel with the outline of the neurons drawn for clarity. Arrowhead in (A) indicates a rare dendritic GFP-Cnn-P1 puncta within a proximal dendritic bubble, while arrowheads in (B) indicate more frequent GFP-Cnn-P1 accumulations within distal dendritic bubbles. (C) Confocal images show the somas and some proximal dendrites of sensory neurons within the dorsal cluster from a 3rd instar *cnn* mutant larva expressing endogenously-tagged γ-tubulin-GFP and immunostained for GFP (green), HRP (greyscale)

*Figure 4 continued on next page*

*Figure 4 continued*

and Cnn (magenta). Images in (**C'**) and (**C''**) show the separate channels for the class I soma and the presumptive basal body of an es neuron, respectively.

The online version of this article includes the following video and figure supplement(s) for figure 4:

**Figure supplement 1.** Cnn-P1 is not strongly associated with the somatic Golgi or Golgi outposts of sensory neurons, while Cnn-P2 is expressed in ensheathing glia.

**Figure 4—video 1.** Endogenously-tagged GFP-Cnn-P1 localises as expected to centrosomes within *Drosophila* syncytial embryos.

https://elifesciences.org/articles/58943#fig4video1

catastrophe or sites of microtubule nucleation, the position of emerging EB1-GFP comets has been routinely used in *Drosophila* neurons as a proxy for nucleation sites (*Nguyen et al., 2014*; *Ori-McKenney et al., 2012*; *Weiner et al., 2020*; *Yalgin et al., 2015*; *Zhou et al., 2014*). It is generally considered that comets repeatedly emerging from the same location are likely to represent nucleation sites (*Ori-McKenney et al., 2012*; *Zhou et al., 2014*). We therefore imaged the soma of class I neurons expressing EB1-GFP and the Golgi marker ManII-mCherry and manually tracked each EB1-GFP comet (*Figure 6A*; *Figure 6—Video 1*; the circle of each track represents the latest position of the EB1 comet). Comets could be observed originating at both Golgi stacks and non-Golgi sites. They emerged at different Golgi stacks within the same cell, and sequentially from the same Golgi stack, suggestive of nucleation. Intriguingly, many Golgi-derived comets appeared to grow towards the axon. Measuring the initial angle of each comet's growth relative to the axon entry site (indicated in *Figure 6A*) showed that there was a strong bias for initial growth towards the axon (163 comets analysed from 59 Golgi stacks from seven neurons, p<0.001), although some comets did grow away from the axon and thus towards dendrites (*Figure 6B*). While it is possible that the direction of comet growth can be influenced by comets growing into the soma from the dendrites, which will tend to grow towards the axon, we minimised this effect by quantifying only those comets that originate from Golgi stacks. The comets we analysed are therefore more likely to represent true growth events from the Golgi rather than catastrophe and regrowth of microtubules that originated from dendrites.

We also noticed that the direction of initial growth of each comet that emerged from the same Golgi stack was similar, irrespective of their angle from the axon. We therefore generated and plotted normalised resultant vectors (final vector position from a (0,0) origin represented by a blue circle) for the comets from each Golgi stack that produced two or more comets (*Figure 6C*; see Materials and methods). The angle between the positive Y-axis and a line connecting (0,0) and a given circle on the graph is representative of the overall comet angle from the axon; the distance (d) of each circle from (0,0) is representative of the similarity of comet angles (d = 1 if all angles are the same; d = 0 if all angles are evenly distributed). 77% of the circles were in the upper quadrants (*Figure 6C*; p<0.001), again showing that there was a preference for comets to grow initially towards the axon. Moreover, there was a bias for resultant vector lengths to be large, as compared to randomly generated data (*Figure 6D*; p<0.001). We conclude that comets from a particular Golgi stack emerge within a small angle with respect to each other and with a directional preference towards the axon. While not definitive, the similarity in the direction of comets emerging repeatedly from the same Golgi stack is indicative of consecutive microtubule nucleation events.

## A temperature-based microtubule nucleation assay suggests that microtubules are nucleated from the somatic Golgi in class I da neurons

The origin of EB1-GFP comets is not a perfect proxy for microtubule nucleation, as EB1-GFP comets also appear during the regrowth of partially depolymerised microtubules. We therefore performed a microtubule nucleation assay using a temperature-control device (CherryTemp, Cherry Biotech) to cool samples rapidly to 5°C and then re-heat them to 20°C during continuous imaging of the sample. Cooling typically causes depolymerisation of dynamic microtubules and warming causes their regrowth. Cooling-warming microtubule nucleation assays have previously been performed in various systems, including *Drosophila* embryos (*Hayward et al., 2014*), *Drosophila* S2 cells (*Bucciarelli et al., 2009*), and in mammalian cells (*Torosantucci et al., 2008*). While populations of cold-stable microtubules have been identified in mammalian neurons, cold stability is thought to be

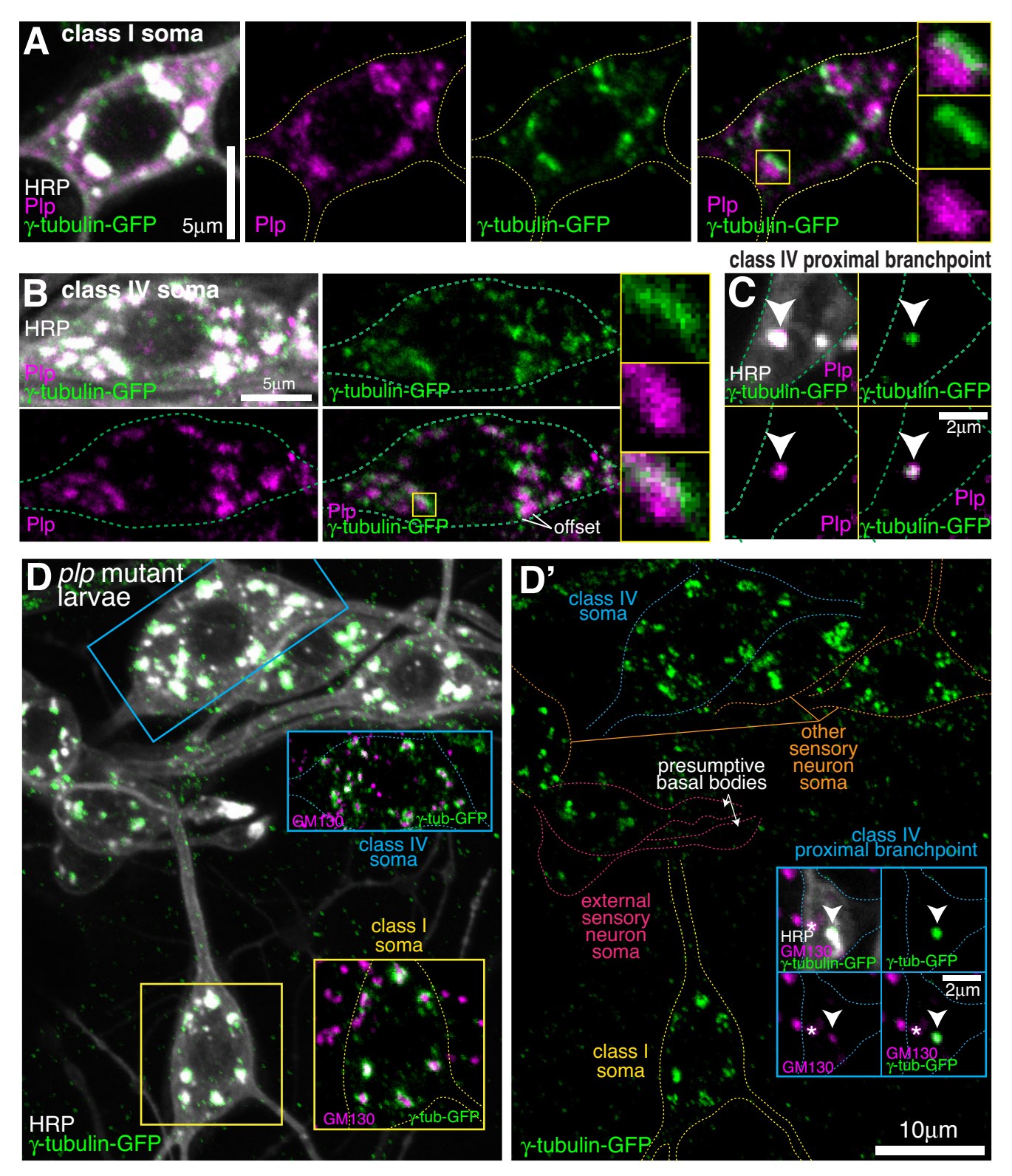

**Figure 5.** Endogenous Plp localises to Golgi but is dispensable for γ-tubulin-GFP recruitment. (A–C) Confocal images show the somas of a class I (A) or a class IV (B) neuron and a proximal branchpoint of a class IV neuron (C) from 3rd instar larvae expressing γ-tubulin-GFP fixed and immunostained for GFP (green), Plp (magenta) and HRP (greyscale). Arrowhead in (C) indicates a γ-tubulin-GFP-positive Golgi outpost that contains Plp. (D) Confocal images show the somas and some proximal dendrites of sensory neurons within the dorsal cluster: class I (yellow), class IV (blue), es (pink) and other

*Figure 5 continued on next page*

*Figure 5 continued*

(orange), from a *plp* mutant 3rd instar larva expressing endogenously-tagged γ-tubulin-GFP and immunostained for GFP (green), HRP (greyscale) and GM130 (magenta). Left panel (D) shows an overlay of the GFP and HRP signal, with insets showing overlays for different neurons, as indicated; right panel (D′) shows the GFP channel with coloured outlines of the different neurons drawn for clarity, with insets showing separate channels for a proximal branchpoint.

The online version of this article includes the following figure supplement(s) for figure 5:

**Figure supplement 1.** Plp localises to the somatic Golgi, Golgi outposts, basal bodies and class I -specific dendritic bubbles.

induced by binding to MAP6 proteins, which are specific to vertebrates (*Bosc et al., 2003*; *Delphin et al., 2012*). We therefore expected that cooling would result in the depolymerisation of at least the dynamic microtubules within *Drosophila* neurons, allowing us to correlate the position of new comet growth with nucleation sites.

During cooling-warming cycles we observed no obvious effect on the distribution of the somatic Golgi stacks (*Figure 6E*; *Figure 6—Video 2*), although thermal-fluctuation-induced movement of the glass coverslip made it difficult to follow the first few timepoints after cooling or warming. When the appropriate focal plane was reached shortly after warming, we could observe EB1-GFP comets emerging from the ManII-mCherry-labelled Golgi structures (green-labelled comets, *Figure 6E*; *Figure 6—Video 2*), suggestive of Golgi-located microtubule nucleation events. Within 20 s of warming in *Figure 6—Video 2*, four comets emerged from Golgi structures (green-labelled comets, *Figure 6E*; *Figure 6—Video 2*), while two comets emerged away from the Golgi (purple-labelled comets, *Figure 6E*; *Figure 6—Video 2*). During the 2 min and 12 s between warming and the end of the movie, a total of 8 comets emerged from the Golgi with a total of five emerging from non-Golgi sites. The initial emergence of non-Golgi comets is not surprising, given that HAUS, the mammalian homologue of *Drosophila* Augmin that recruits γ-TuRCs to the sides of pre-existing microtubules, is required for microtubule nucleation events within the soma of mammalian neurons in culture (*Sánchez-Huertas et al., 2016*). It may also be, however, that the non-Golgi comets grew initially from Golgi stacks in a different focal plane or that these comets represented the re-growth of microtubules that were not fully depolymerised. Indeed, while EB1-GFP comets disappeared rapidly on cooling, suggesting that dynamic microtubules quickly lose their GTP-tubulin cap and thus depolymerise, it is possible that a proportion of microtubules within the soma are cold stable and therefore won't depolymerise. Presumably, most of these stable microtubules would remain stable, but it is possible that some would start to grow and thus contribute to the EB1-GFP comets that we observe on warming.

In our opinion, the best evidence for microtubule nucleation occurring at the somatic Golgi comes from the observation that several comets emerged from Golgi stacks relatively late after warming. For example, in *Figure 6—Video 2* four comets emerged from Golgi stacks at least 50 s post warming. While some of these late comets emerged from a Golgi stack that had generated a comet immediately after warming, others emerged from Golgi stacks that had not generated a comet immediately after warming. Most importantly, the direction of all late emerging comets suggests that they were not simply generated by catastrophe-rescue of a microtubule that had grown immediately after warming. While it is impossible to rule out that these late emerging comets could have been generated by re-growing microtubules that were originally out of focus, the data strongly suggests that they instead represent genuine nucleation events from the Golgi .

## Growing microtubules within the soma are guided towards the axon while being excluded from entering dendrites in a Kinesin-2-dependent manner

We next wanted to determine the fate of growing microtubules within the soma. We therefore imaged and tracked EB1-GFP comets within the soma and proximal axons and dendrites of 13 class I da neurons (*Figure 7A*; *Figure 7—Video 1*). These neurons have at least two primary dendrites but only a single axon; however, comets that initiated within the soma often grew into the axon, while few entered dendrites (*Figure 7—Video 1*). We found that this was due to two major factors. The first was that a higher proportion of comets reached the axon entry site: of the 666 comets that had initiated within the soma across all movies 104 approached the entrance to the axon (15.6%),

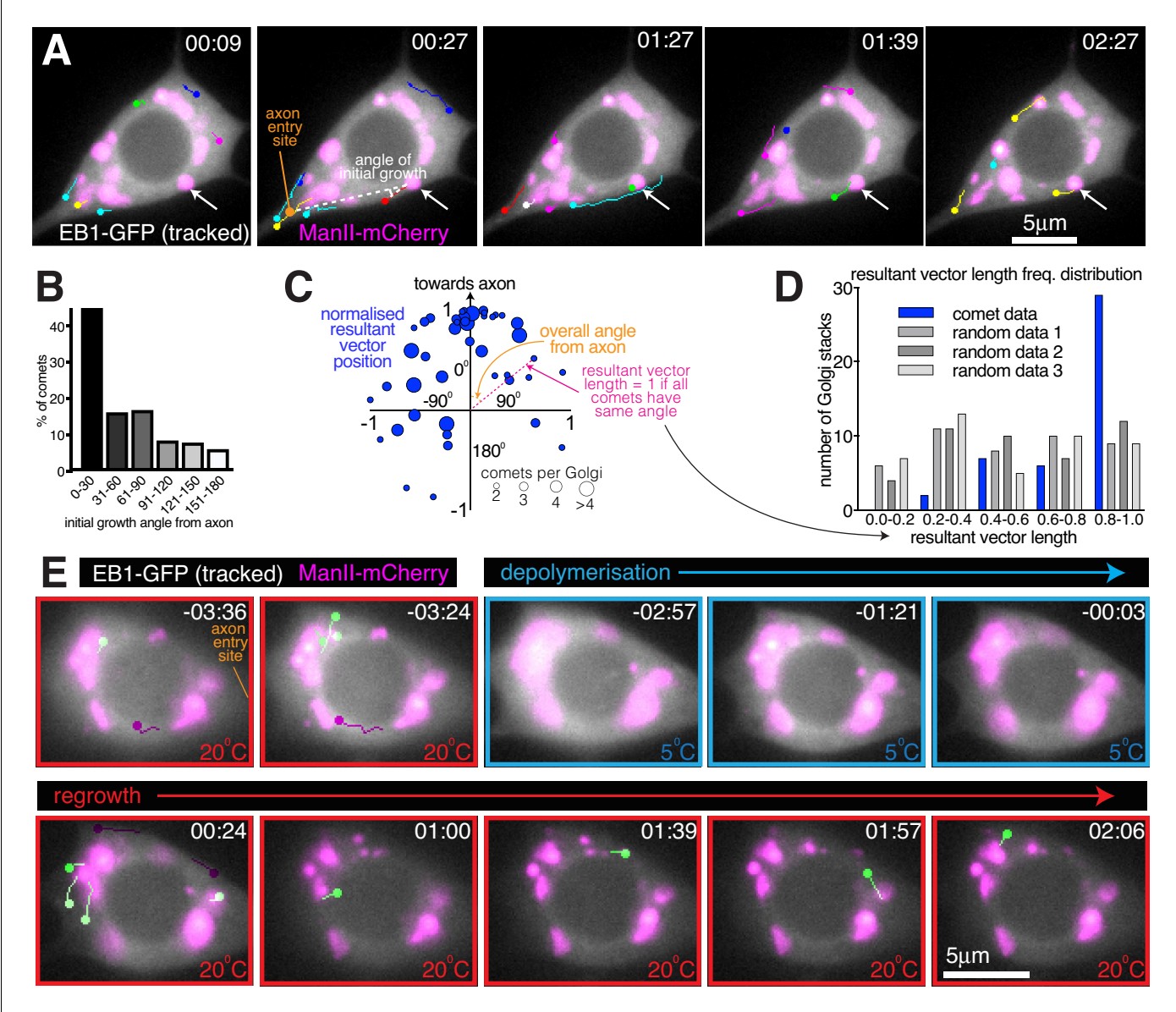

**Figure 6.** Somatic Golgi stacks nucleate microtubules. (**A**) Widefield fluorescent images from a movie showing the soma of a class I da neuron expressing EB1-GFP (greyscale) and ManII-mCherry (magenta). Manually assigned multi-colour tracks of EB1-GFP comets were drawn over each image (filled circle = last location of comet). Arrow indicates a Golgi stack from which sequential EB1-GFP comets emerge; dotted lines show an example of an angle measurement of a comet's initial growth relative to the axon entry point (orange circle). Time in min:s from the start of the movie is shown. (**B**) Graph shows a frequency distribution of the initial growth angle (before turning) relative to the axon entry site (as indicated in (**A**)) of EB1-GFP comets that emerged from Golgi stacks. Negative angles were made positive so as not to distinguish between comets growing to the left or right of the axon. 163 comets from 59 Golgi stacks from seven neurons were analysed. (**C**) Scatter graph shows the position of the normalised resultant comet vectors (see methods) for each of 44 Golgi stacks from seven neurons. Circle position is representative of the average angle from the axon and how similar the angles were (as indicated). Circle size reflects the number of comets analysed per Golgi stack (as indicated below). (**D**) Graph shows a frequency distribution of the resultant vector lengths for the comet data (blue) and for 3 sets of data produced using randomly generated angles (grey shades). (**E**) Widefield fluorescent images from a movie showing the soma of class I da neuron expressing UAS-EB1-GFP (greyscale) and UAS-ManII-mCherry (magenta). The sample was subjected to a warm-cool-warm temperature cycle to induce depolymerisation (5 °C; blue outline) and then repolymerisation (20 °C; red outline) of dynamic microtubules. Tracks were manually colour coded: green = comets emerging from Golgi stacks; purple = comets emerging from elsewhere. Time in min:s from the 5°C to 20°C transition is shown.

The online version of this article includes the following video and source data for figure 6:

**Source data 1.** Calculating the angle of microtubule growth from Golgi stacks.

**Figure 6—video 1.** EB1-comets emerge from somatic Golgi stacks and initially grow preferentially towards the axon.

*Figure 6 continued on next page*

*Figure 6 continued*

https://elifesciences.org/articles/58943#fig6video1

**Figure 6—video 2.** The somatic Golgi stacks nucleate microtubules.

https://elifesciences.org/articles/58943#fig6video2

while only 47 approached the entrance of a dendrite (7.1%) (*Figure 7C*; p<0.001); the remaining 77.3% of comets terminated within the soma away from the axon or dendrites. The second important factor was that when comets arrived at either the axon entry site or a dendritic entry site, they had more chance of entering the axon: of the 104 comets that approached the entrance to an axon across all movies 56 entered (53.8%), while of the 47 comets that approached the entrance to a dendrite only 13 entered (27.7%) (*Figure 7D*; p<0.001). Overall, 8.4% of all comets that originated in the soma entered the axon while only 2.0% entered a dendrite. In summary, growing microtubule plus ends within the soma preferentially reach the entrance to the axon and can readily enter, while the few that reach dendrites are normally excluded from entering.

While asymmetric nucleation from the somatic Golgi likely contributes to more microtubule plus ends reaching the axon (*Figure 6*) we also observed several occasions where microtubules turned towards the axon (*Figure 7A*; *Figure 7—Video 1*). Microtubule "collision resolution" events have been observed previously within dendritic branchpoints of da neurons, where the microtubules turn towards the soma along stable microtubules (*Mattie et al., 2010*; *Weiner et al., 2016*). This depends upon Kinesin-2, which is a heterotrimeric plus-end-directed motor whose regulatory subunit, Kap3, interacts with EB1 via APC (*Mattie et al., 2010*). It has been proposed that plus-end-associated Kinesin-2 guides growing microtubules along and towards the plus end of so-called 'rail' microtubules, as this can be recapitulated *in vitro* (*Chen et al., 2014*; *Doodhi et al., 2014*). When we depleted Kap3 from class I da neurons by RNAi there was a dramatic reduction in the frequency of microtubule turning events within the soma. In control cells, of the 257 comets across all movies that grew for more than 2 µm within the soma and that did not travel along the cell cortex, 165 displayed turning behaviour (64.2%), while across 9 Kap3 RNAi cells only 35 of the 386 qualifying comets displayed turning behaviour (9.1%) (*Figure 7A,B,E*; *Figure 7—Videos 1* and *2*; p<0.001). Comets in Kap3 RNAi cells tended to change direction only when travelling along the cell cortex or after collision with the nuclear envelope or cell cortex (*Figure 7B*; *Figure 7—Video 2*). As a result of reduced turning in Kap3 RNAi neurons, a lower proportion of comets reached the axon entry site and a higher proportion reached dendritic entry sites, compared to control cells: of the 1058 comets that originated within the soma of Kap3 RNAi cells, 97 approached the axon entry site (9.2%, compared to 15.6% in control neurons, p<0.001) and 114 approached a dendritic entry site (10.8%, compared to 7.1% in control neurons, p=0.0098) (*Figure 7C*). These data suggest that Kinesin-2 guides growing plus ends along pre-existing microtubules towards the axon and away from dendrites within the soma.

Comets that did arrive at the axon entry site could still readily enter the axon after Kap3 RNAi (*Figure 7D*); there was actually a ~ 1.3 fold increase compared to controls in the proportion of comets entering the axon (72.2% versus 53.5%, p=0.007). There is no requirement for Kinesin-2 for microtubule growth per se, which is presumably why microtubules can still enter once they reach the axon. It is unclear why entry is increased, but it is possible that Kinesin-2 may limit plus-end growth in general, although this purely speculative. Importantly, increased entry of growing microtubules into the axon has no major effect on microtubule polarity, as axons normally contain predominantly plus-end-out microtubules.

Most significantly, the proportion of comets that entered dendrites after Kap3 RNAi increased ~2 fold compared to controls, a much higher increase than that seen for axons. Of the 114 comets across nine movies that approached the entrance to a dendrite in Kap3 RNAi neurons, 65 entered (57.0%, compared to 27.7% in control neurons; p<0.001; *Figure 7D*). This affect was even more striking when knocking down a different Kinesin-2 component, Klp64D, where 56 of the 80 comets (across 10 movies) that approached a dendrite could enter (70.0%; p<0.001; *Figure 7—Video 3*). Increased entry of comets into dendrites contributed to, if not caused, a dramatic increase in the proportion of anterograde comets within proximal primary dendrites (prior to any dendritic branches): in control neurons, the vast majority of comets that we could observe in all proximal

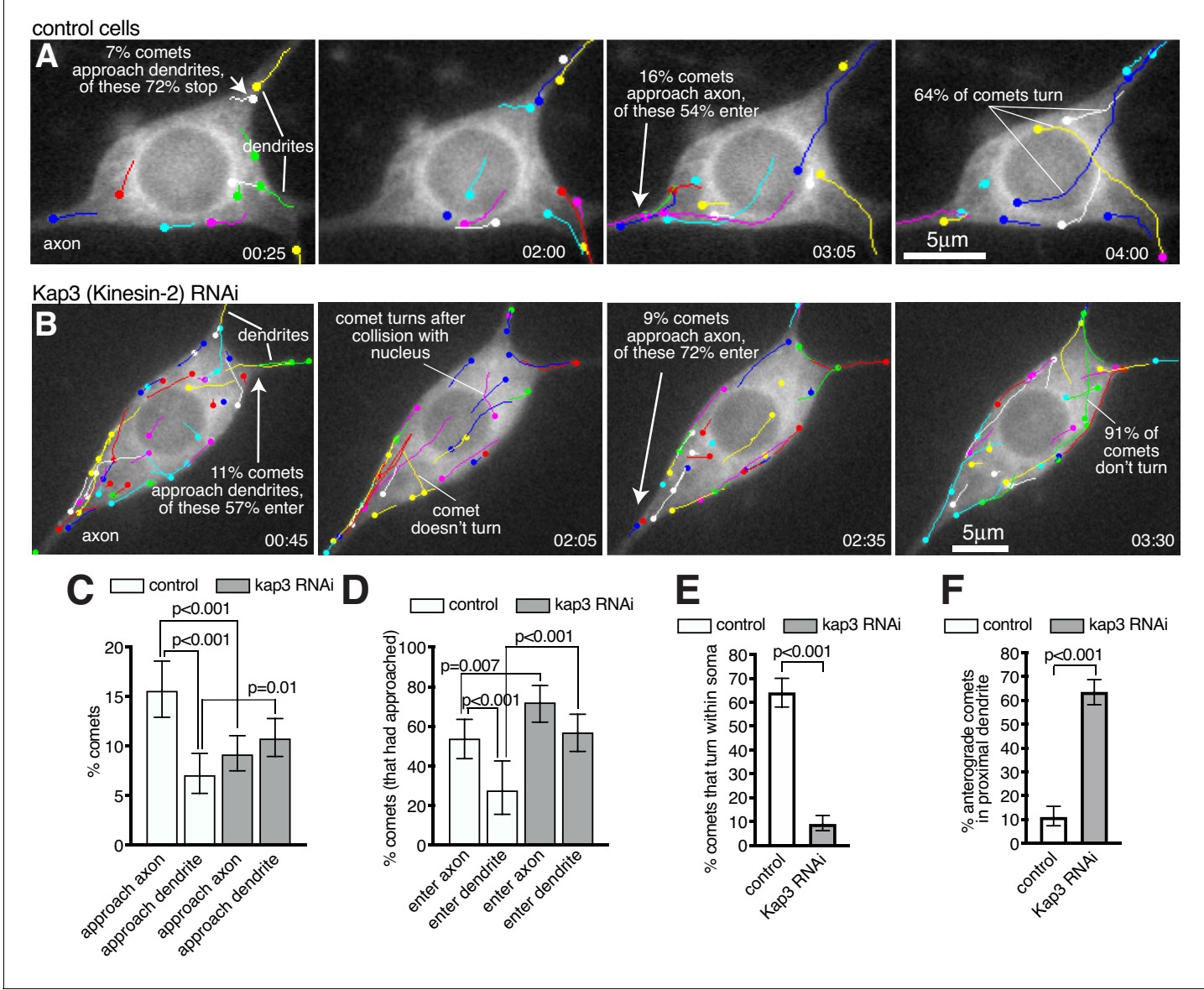

**Figure 7.** Microtubules within the soma grow preferentially towards the axon and are excluded from entering dendrites in a Kinesin-2-dependent manner. (**A,B**) Widefield fluorescent images from movies showing the somas of class I da neurons expressing UAS-EB1-GFP and either γ-tubulin-37C RNAi (control) (**A**) or UAS-Kap3 RNAi (**B**). Manually assigned multi-colour tracks of EB1-GFP comets are drawn over each image (filled circle = last location of comet). Time in min:s from the start of the movie is shown. (**C**) Graph shows the % of comets that approach either the axon (green) or the dendrites (magenta) in either control (666 comets analysed across 13 movies) or Kinesin-2 RNAi (1058 comets analysed across nine movies) class I da neuron somas, as indicated. (**D**) Graph shows the % of comets that enter the axon (green) or the dendrites (magenta) as a proportion of those that had approached the axon in either control or Kinesin-2 RNAi class I da neuron somas, as indicated. (**E**) Graph shows the % of applicable comets that display turning events within either control (n = 257 comets across 13 movies) or Kinesin-2 RNAi (n = 386 comets across nine movies) class I da neuron somas, as indicated. (**F**) Graph shows the % of comets that are anterograde in the proximal primary dendrite (before any branches) in either control (n = 252 comets across 13 movies) or Kinesin-2 RNAi (n = 338 comets across nine movies) class I da neurons, as indicated. Error bars in (**C–F**) show the 95% confidence intervals.

The online version of this article includes the following video and source data for figure 7:

**Source data 1.** Calculating percentages of comets aproaching and entering axons and dendrites, turning events, and proximal dendrite polarity in control neurons.

**Source data 2.** Calculating percentages of comets aproaching and entering axons and dendrites, turning events, and proximal dendrite polarity in Kap3 RNAi neurons.

**Source data 3.** Calculating percentages of comets entering dendrites in Klp64D RNAi neurons.

**Figure 7—video 1.** Microtubules turn towards the axon and are excluded from entering dendrites.

*Figure 7 continued on next page*

*Figure 7 continued*

https://elifesciences.org/articles/58943#fig7video1

**Figure 7—video 2.** Kinesin-2 is required for growing microtubules to turn within the soma and to be excluded from entering dendrites.

https://elifesciences.org/articles/58943#fig7video2

**Figure 7—video 3.** Knockdown of Klp64D supports the conclusion that Kinesin2 is required for growing microtubules to be excluded from entering dendrites.

https://elifesciences.org/articles/58943#fig7video3

dendrites (88.9%) were retrograde (minus-end-out microtubules), while in Kap3 RNAi neurons the majority (63.6%) were anterograde (plus-end-out microtubules) (*Figure 7F*; p<0.001). An explanation for Kinesin-2 being required to prevent growing microtubules from entering dendrites may be that these microtubules need to grow past dendritic microtubules of opposite polarity. If plus-end-associated Kinesin-2 engages with these oppositely polarised microtubules it would presumably create a backward stalling force when trying to 'walk' towards the plus end of the dendritic microtubule (see discussion for more detail).

We conclude that Kinesin-2 is required to guide growing microtubule plus ends within the soma towards the axon and to prevent them from entering dendrites, which is essential in order to maintain minus-end-out microtubule polarity within proximal dendrites.

## A model for microtubule regulation within the neuronal soma

Based on our observations, and the findings from previous studies (*Chen et al., 2014*; *Doodhi et al., 2014*; *Mattie et al., 2010*; *Weiner et al., 2016*), we propose a model to explain how microtubules are generated and regulated within the soma of da neurons, which also helps explain how microtubule polarity is promoted in axons and maintained within proximal dendrites (*Figure 8*). In this model, γ-TuRCs are localised to the cis-Golgi and microtubules are nucleated preferentially towards the axon, generating a polarised microtubule network within the soma. Plus-end-associated Kinesin-2 guides the growing plus ends of the nucleated microtubules along and towards the plus ends of the polarised microtubule network, and thus guides them towards the axon and away from dendrites (*Figure 8A,C*). Microtubules that happen to grow towards a dendrite are excluded from entering when the plus-end-associated Kinesin-2 engages with the shaft of a dendritic microtubule of opposite polarity and thus exerts a backward stalling force on the growing plus end (*Figure 8B, C*). Importantly, this model explains how minus-end-out microtubule polarity is maintained within proximal dendrites.

## Discussion

Understanding of how neuronal microtubules are generated and organised is a matter of much interest and importance. In this paper, we have used *Drosophila* da neurons as an *in vivo* model system to address these questions. Our results have led us to propose a model that describes how microtubules are generated and organised within the soma, which also impacts the microtubule populations within proximal axons and dendrites. This model comprises multiple elements. Firstly, the model states that microtubules are nucleated asymmetrically from the somatic Golgi, growing initially with a preference towards the axon and away from dendrites. Secondly, plus-end-associated Kinesin-2 helps guide these growing microtubules towards the axon and away from dendrites along a polarised network of microtubules within the soma. Thirdly, plus-end-associated Kinesin-2 also helps prevent any microtubules that do approach the entrance to a dendrite from entering. We propose that a combination of these mechanisms is essential to maintain minus-end-out polarity in proximal dendrites, a key feature that distinguishes dendrites from axons.

We have shown that endogenously tagged γ-tubulin23C-GFP, which is a proxy for the major catalysts of microtubule nucleation, γ-TuRCs, localises predominantly to the Golgi stacks within the soma of da neurons, and to a few Golgi outposts within the proximal branches of class IV da neurons. Our data show that the ability of γ-tubulin to localise to these Golgi structures does not depend upon either Cnn or Plp, both of which have been implicated in recruiting γ-TuRCs to Golgi outposts within dendrites of class I and class IV da neurons, respectively (*Ori-McKenney et al., 2012*; *Yalgin et al.,*

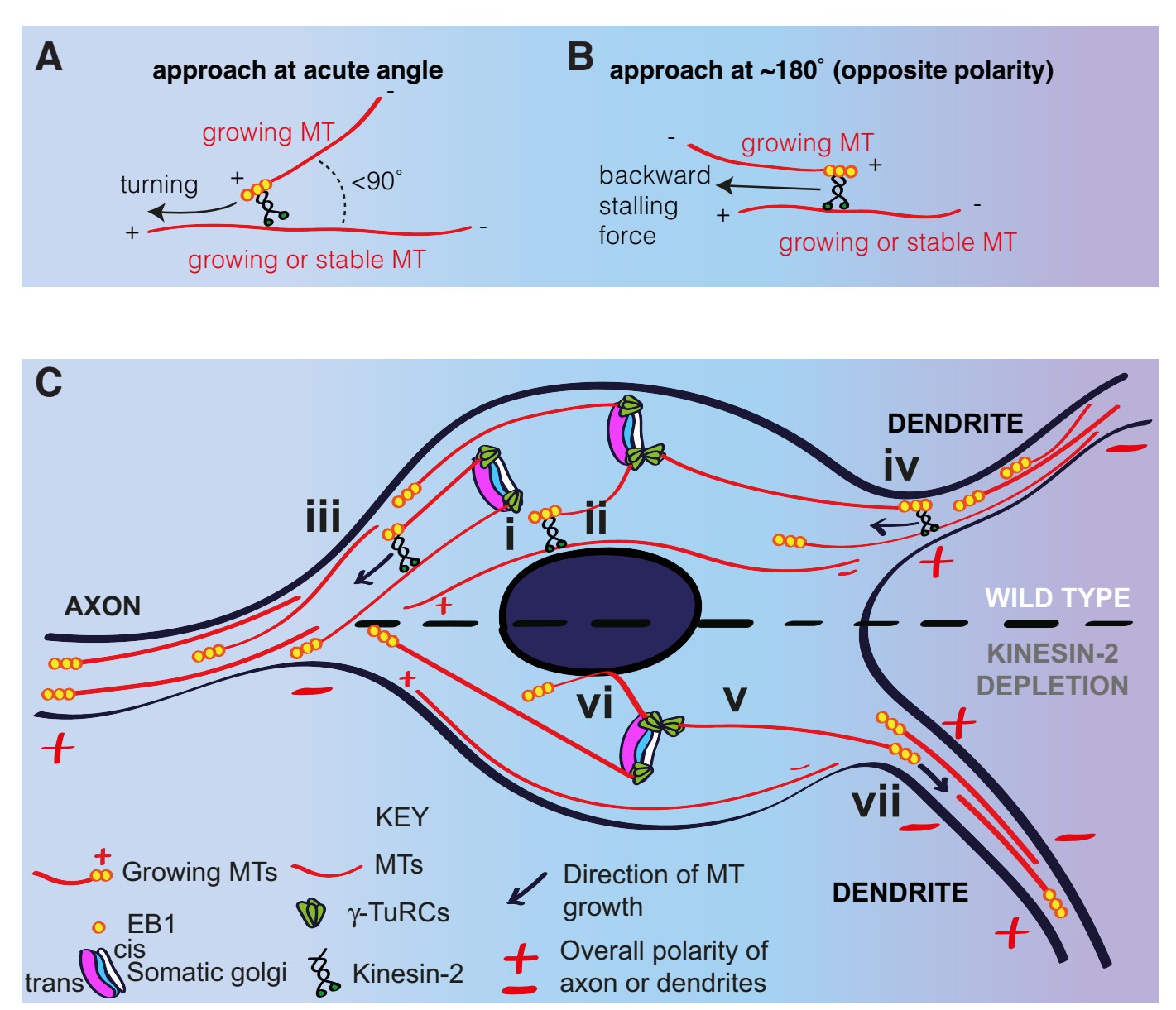

**Figure 8.** A model for microtubule nucleation and organisation within the soma of da neurons. (**A,B**) Diagrams showing how plus-end-associated Kinesin-2 is predicted to induce turning of a growing microtubule along another microtubule when it approaches the other microtubule at an acute angle (**A**), or plus-end stalling when the growing microtubule approaches the other microtubules at an angle of ~180° (**B**). (**C**) A model depicting microtubule nucleation and growth within the soma of wild type (upper section) or Kinesin-2 depleted (lower section) da neurons. γ-TuRCs localise to the cis-Golgi (white) and nucleate microtubules preferentially towards the axon (i), creating a polarised microtubule network. In wild type neurons, plus-end-associated Kinesin-2 guides growing microtubules along this network towards the axon (ii). Growing microtubules readily enter axons (iii) but are excluded from entering dendrites (iv), because plus-end-associated Kinesin-2 engages with microtubules of opposite polarity, exerting a backward stalling force on the growing microtubule. In Kinesin-2 depleted neurons, microtubules nucleated from the somatic Golgi grow in straight lines (v) until contact with the nuclear envelope (vi) or the cell cortex (not depicted). Microtubules that approach dendrites can now readily enter (vii) resulting in a loss of overall minus-end-out microtubule polarity.

*2015*). Thus, γ-tubulin, presumably in the form of γ-TuRCs, must be recruited to the Golgi by another tethering protein, perhaps by one of the large coiled-coil 'Golgin' proteins that project from specific parts of the Golgi (*Munro, 2011*).

After showing that γ-tubulin-GFP localised to the somatic Golgi, we tracked EB1-GFP comets from the Golgi and found that they grew with an initial directional preference towards the axon,

suggestive of asymmetric nucleation. This analysis was more robust under normal conditions than after cooling-warming cycles, because more comets could be analysed (it was difficult to keep the imaging plane consistent during temperature shifting, due to temperature-induced glass expansion/contraction). Thus, while an equal number of Golgi-derived comets (4) grew towards and away from the axon after warming in *Figure 6—Video 2*, we believe this perceived lack of asymmetry is most likely due to chance, especially as a fraction of comets can grow in more random directions even under normal conditions (*Figure 6C*). We therefore propose that there is a genuine preference for growth of Golgi-derived microtubules towards the axon. This may be explained, at least in part, by the asymmetric localisation of γ-tubulin at the somatic Golgi stacks. Using markers of cis- and trans-Golgi, we have found that γ-tubulin localises preferentially to the cis-Golgi. This is also true in mammalian cells, where the somatic Golgi acts as an asymmetric MTOC in various cell types, such as migrating fibroblasts and different types of cultured mammalian cells in interphase (*Rios, 2014*). It has been proposed that asymmetric microtubule nucleation is due to the presence of CLASP proteins at the trans-Golgi, which have been suggested to bind and stabilise the plus ends of microtubules that are nucleated by γ-TuRCs at the cis-Golgi (*Vinogradova et al., 2009*). Whether this occurs in *Drosophila* neurons remains to be explored. *Drosophila* CLASP is known to be a plus-tip protein within axons (*Beaven et al., 2015*) and is required for axon guidance (*Lee et al., 2004*), but to our knowledge a role for CLASP at the Golgi in any *Drosophila* cell type has not yet been reported. In mammalian cells, directional microtubule nucleation is thought to be important for cell motility in fibroblasts (*Efimov et al., 2007*) and for polarised pseudopod formation during invasion of cancer cells (*Wu et al., 2016*). Our data suggest that asymmetric nucleation at the somatic Golgi in *Drosophila* da neurons may establish a polarised microtubule network within the soma and is important for ensuring that plus-end-out microtubules enter axons rather than dendrites.

It is possible that a fraction of comets that originate from the somatic Golgi represent microtubules that were re-growing after partial depolymerisation, where the re-growth event happened to overlap a Golgi stack. However, several pieces of evidence suggest that the majority of the Golgi-derived comets represent nucleation events. First, γ-tubulin accumulates strongly at the somatic Golgi and a high accumulation of γ-TuRCs at a particular site is normally indicative of a nucleation site, such as when γ-TuRCs accumulate at mitotic centrosomes. Second, we have found that comets emerge repeatedly from the Golgi in a similar direction, strongly suggestive of repeated nucleation events. Third, we believe it is unlikely that all of the comets emerging from the Golgi after a cooling-warming cycle represent microtubules that had not fully depolymerised and that just happened to re-grow close to the Golgi. Moreover, comets that emerged from the Golgi a significant amount of time after warming did not emerge in the same direction as those that had grown immediately after warming. This strongly suggests they are not the same microtubule re-growing after depolymerisation but represent new nucleation events. Direct imaging of microtubules could help assess the degree of overall microtubule depolymerisation induced by cooling, although cold-resistant microtubules that may remain after cooling could impede the ability to visualise depolymerisation of dynamic microtubules, especially if the dynamic microtubules had originally grown alongside the stable microtubules. Increasing the cooling time before inducing regrowth may cause the depolymerisation of all microtubules, and this could be tested by using fluorescent markers of microtubules or by fixing and immunostaining the samples. These experiments have limitations, however, as keeping the animal alive for long enough to depolymerise all microtubule may be challenging and fixing and staining experiments do not allow one to observe the same neuron before, during, and after cooling. In summary, when we consider all of our data as a whole, we believe the most parsimonious conclusion is that microtubules are nucleated asymmetrically from the somatic Golgi.

It has previously been shown that centrosomes in *Drosophila* neurons do not act as MTOCs (*Nguyen et al., 2011*), and our observations that EB1-GFP comets do not emerge radially from a single source support this conclusion. In other cell types the somatic Golgi can 'compete' with centrosomes for the ability to organise microtubules, where decreasing nucleation from one organelle increases nucleation at the other (*Gavilan et al., 2018*; *Ríos et al., 2004*; *Wu et al., 2016*). It is therefore possible that the lack of microtubule nucleation at centrosomes in *Drosophila* neurons enables microtubules to be nucleated at the Golgi. In mammalian neurons the centrosome is deactivated after the first few days in culture (*Stiess et al., 2010*). An intriguing possibility is that this deactivation allows the Golgi to organise microtubules, which may be necessary to ensure correct

microtubule polarity during later neuronal development, although nucleation from the Golgi in mammalian neurons has yet to be reported.

Our analysis was carried out in third instar larvae where the neurons are fully developed and functional. Axons have already extended and made connections in the central nervous system and dendritic arbors are largely established. It is thus somewhat surprising that microtubules continue to be nucleated within the soma. It is possible that somatic microtubules are necessary for transport of cargos to axonal and dendritic entry sites. Given that Golgi-derived microtubules grow preferentially towards the axon and that somatic Golgi stacks are distributed throughout the soma, there is likely an overall microtubule polarity within the soma itself, with more plus ends pointing towards the axon. Even small biases in the orientation of microtubules within a network can be important for the polarised distribution of components, such as in *Drosophila* oocytes (*Zimyanin et al., 2008*). A similar bias within neuronal soma could be important for directing molecules into axons and dendrites. Perhaps this asymmetric microtubule network within the soma, as well as the microtubules within the axon, need constant renewal. There is evidence that molecular motors can damage microtubules (*Triclin et al., 2018*) and so the large amount of motor-driven transport that occurs within neurons may necessitate microtubule renewal from the soma.

A key feature of our model is the action of plus-end-associated Kinesin-2. Our Kap3 RNAi data suggest that Kinesin-2 is required to guide growing microtubules along a polarised microtubule network within the soma towards the axon and away from dendrites, although future experiments with dual-colour imaging of microtubules and EB1-GFP comets will help support this conclusion. Our data also suggest that plus-end-associated Kinesin-2 prevents outward growing plus ends of somatic microtubules from entering dendrites. Within the soma, we envisage that when a somatic microtubule approaches another somatic microtubule at an acute angle (<90°), it can readily turn and be guided along that microtubule (*Figure 8A,C*). Kinesin-2 mediated "collision resolution" events occur at dendritic branch points in class I da neurons to ensure growing microtubules turn towards the soma (*Weiner et al., 2016*) and turning events have been recapitulated *in vitro* (*Chen et al., 2014*; *Doodhi et al., 2014*). One study showed that the *in vitro* turning events also occured at obtuse angles (>90°), where the growing microtubule buckled to allow the plus end to remain in contact with the 'rail' microtubule. This could presumably also occur within the soma of da neurons. The frequency of guidance events, however, decreases significantly as the angle increases (*Doodhi et al., 2014*). In contrast to the microtubule collision events that will occur with variable angles within the neuronal soma, when outward growing somatic microtubules try to grow into a dendrite they will always encounter dendritic microtubules at angles close to 180° (*Figure 8B,C*), because most dendritic microtubules are oriented minus-end-out and also because of the limited space available at dendrite entry points. Kinesin-2 motors on the plus end of the outward growing somatic microtubules could engage with the shafts of the minus-end-out dendritic microtubules and generate a backwards force when attempting to 'walk' towards the plus end of the dendritic microtubule i.e. in a direction opposite to which the plus end of the somatic microtubule is attempting to grow. This backward force would presumably lead to stalling of the plus end and thus depolymerisation of the microtubule. It is also possible that Kinesin-2 motors slide the plus end along the dendritic microtubule back into the soma, but this sliding action requires the growing microtubule to buckle and thus requires sufficient space (*Doodhi et al., 2014*). Thus, buckling may be impeded at the narrow dendrite entry site anddepolymerisation favoured. Kinesin-2 motors should be sufficient to induce stalling of microtubule growth, because growing microtubules generate a force of ~3–4 pN (*Dogterom and Yurke, 1997*), while a single Kinesin-2 motor generates a force of ~5 pN (*Schroeder et al., 2012*). Depolymerisation upon stalling could be equivalent to how depolymerisation is induced when microtubules grow against immovable objects (*Janson et al., 2003*). Importantly, this model elegantly explains why microtubules from the soma are prevented from growing into the dendrites while dendritic microtubules can readily grow into the soma. Other models, such as if a high concentration of a microtubule depolymerase was present at dendritic entry sites, could not distinguish between these two events. In contrast, our model predicts that dendritic microtubules can grow readily into the soma because they do not frequently encounter a microtubule with opposite polarity, except for the occasional somatic microtubule attempting to grow into a dendrite.

Our model explains how minus-end-out microtubule polarity is maintained within proximal dendrites, but what is the origin of minus-end-out microtubules? These microtubules could have been

pushed out from the soma minus end first by motor-based sliding, a mechanism known to promote initial axon outgrowth in cultured *Drosophila* neurons (*Lu et al., 2013*; *Lu et al., 2015*). Alternatively, or in conjunction with this, they could have been nucleated at sites within dendrites and then have grown back towards the soma. Retrograde microtubule growth events do occur within the dendrites of da neurons, and some of these growth events originate from Golgi outposts (*Ori-McKenney et al., 2012*; *Yalgin et al., 2015*; *Zhou et al., 2014*) and from early endosomes at branchpoints (*Weiner et al., 2020*). It was shown that γ-tubulin co-localises with early endosome markers at class I branchpoints and that this localisation relies on Wnt signalling proteins (*Weiner et al., 2020*). Our observation that endogenously-tagged γ-tubulin localises to branchpoints supports this conclusion. Moreover, it was recently shown that an endosome-based MTOC tracks the growing dendritic growth cone within *C. elegans* PVD neurons and nucleates microtubules that travel back towards the soma (*Liang et al., 2020*). The authors also suggest that a similar process may occur in newly developing *Drosophila* class I neurons. We also observe γ-tubulin within dendritic bubbles and dendritic stretches, but whether γ-tubulin is also recruited to these regions by endosomes, or by cytosolic accumulations of γ-TuRC tethering proteins such as Cnn or Plp, will need to be addressed in future studies. While our data suggest that γ-TuRCs are absent from the majority of Golgi outposts, they appear to be present at a few proximal Golgi outposts within class IV neurons. Moreover, microtubule growth events from Golgi outposts could be independent of γ-TuRCs; other non-γ-TuRC microtubule binding proteins can promote microtubule nucleation *in vitro* (*Roostalu and Surrey, 2017*), and ectopic Golgi outposts within the axons of nudE mutants promote microtubule growth but not via γ-TuRCs (*Arthur et al., 2015*; *Yang and Wildonger, 2020*). Thus, there are several potential mechanisms to generate minus-end-out microtubules within dendrites, that will need to be explored further in future.

It will be interesting in future to examine whether the plus ends of somatic microtubules can grow into dendrites more readily during early neuronal development, when there may be fewer minus-end-out microtubules within dendrites. If true, there must come a developmental point at which the balance shifts and microtubules are prevented from growing into dendrites. This shift could occur in several different ways, including upregulating the number of retrograde microtubules nucleated within dendrites, upregulating Kinesin-2, or changing the direction of microtubule nucleation within the soma. These are all interesting possibilities that are now open for investigation in the future.

## Materials and methods

### Key resources table

| Reagent type (species) or resource | Designation | Source or reference | Identifiers | Additional information |
|---|---|---|---|---|
| Strain, strain background (*Escherichia coli*) | 5-alpha competent *E. coli* | NEB | C2987I | High Efficiency chemically competent |
| Genetic reagent (*Drosophila melanogaster*) | cnn$^{f04547}$ | Exelixis at Harvard Medical School | Exelixis:f04547; Flybase:FBst1019233 | FlyBase symbol: cnn$^{f04547}$ ; Stock discontinued at the Exelixis and available from Conduit lab upon request. |
| Genetic reagent (*D. melanogaster*) | plp$^5$ | Bloomington *Drosophila* Stock Center | BDSC:9567; FLYB:FBst0009567; RRID:BDSC_9567 | FlyBase symbol: ru1 Plp5 st1/TM6C, Sb1 Tb1 |
| Genetic reagent (*D. melanogaster*) | plp$^{s2172}$ | Bloomington *Drosophila* Stock Center | BDSC:12089; Flybase:FBst0012089; RRID:BDSC_12089 | FlyBase symbol: 'w[1118]; P{w[+mC]=lacW}Plp [s2172]/TM3, Sb[1]' |
| Genetic reagent (*D. melanogaster*) | UAS-mCD8-RFP | Bloomington *Drosophila* Stock Center | BDSC:27392; Flybase:FBti0115769; RRID:BDSC_27392 | FlyBase symbol: 'w[*]; P{w[+mC]=UAS-mCD8.ChRFP}3' |
| Genetic reagent (*D. melanogaster*) | UAS-EB1-GFP | Bloomington *Drosophila* Stock Center | BDSC:35512; Flybase:FBti0141213; RRID:BDSC_35512 | FlyBase symbol: 'w[*]; P{w[+mC]=UAS-EB1-GFP}3' |

*Continued on next page*

Continued

| Reagent type (species) or resource | Designation | Source or reference | Identifiers | Additional information |
|---|---|---|---|---|
| Genetic reagent (*D. melanogaster*) | UAS-ManII-GFP | Bloomington *Drosophila* Stock Center | BDSC:65248; Flybase:FBti018367; RRID:BDSC_65248 | FlyBase symbol: 'w[1118]; P{w[+mC]=UAS-ManII-EGFP}2; TM2/TM6B, Tb[1]' |
| Genetic reagent (*D. melanogaster*) | UAS-ManII-mCherry | *Ori-McKenney et al., 2012* (doi: 10.1016/j.neuron.2012.10.008) | | Fly genotype: 'w[1118]; P{w[+mC]=UAS-ManII-mCherry}/CyO'; gift from Yuh-Nung Jan |
| Genetic reagent (*D. melanogaster*) | ppk-Gal4 | Bloomington *Drosophila* Stock Center | BDSC:32078; Flybase:FBti0127690; RRID:BDSC_32078 | FlyBase symbol: 'w[*]; P{w[+mC]=ppk-GAL4.G}2' |
| Genetic reagent (*D. melanogaster*) | ppk-Gal4 | Bloomington *Drosophila* Stock Center | BDSC:32079; Flybase:FBti013120; RRID:BDSC_32079 | FlyBase symbol: 'w[*]; P{w[+mC]=ppk-GAL4.G}3' |
| Genetic reagent (*D. melanogaster*) | 221-Gal4 | Bloomington *Drosophila* Stock Center | BDSC:26259; Flybase:FBti011433; RRID:BDSC_26259 | FlyBase symbol: 'w[*]; Pin[1]/CyO; P{GawB}221w-' |
| Genetic reagent (*D. melanogaster*) | ppkCD4-tdGFP | Bloomington *Drosophila* Stock Center | BDSC:35842; Flybase:FBti014342; RRID:BDSC_35842 | FlyBase symbol: 'w[1118]; P{w[+mC]=ppk-CD4-tdGFP}1b' |
| Genetic reagent (*D. melanogaster*) | UAS-Dicer 2 | Bloomington *Drosophila* Stock Center | BDSC:24646; Flybase:FBti010143; RRID:BDSC_24646 | FlyBase symbol: P{w[+mC]=UAS-Dcr2.D}1,w[1118] |
| Genetic reagent (*D. melanogaster*) | γ-tubulin37C RNAi; UAS-γ-tubulin37C RNAi | Bloomington *Drosophila* Stock Center | BDSC:32513; Flybase:FBti0132207; RRID:BDSC_32513 | FlyBase symbol: 'y[1] sc[*] v[1] sev[21]; P{y[+t7.7] v[+t1.8]=TRiP.HMS00517}attP2' |
| Genetic reagent (*D. melanogaster*) | Kap3 RNAi; UAS-Kap3 RNAi | Vienna *Drosophila* Resource Center | FBst0466097; VDRC ID:45400 | FlyBase symbol: 'w1118;P{GD7377}v45400' |
| Genetic reagent (*D. melanogaster*) | Klp64D RNAi | Vienna *Drosophila* Resource Center | FBst0466080; VDRC ID:45373 | FlyBase symbol: 'w1118;P{GD7048}v45373' |
| Genetic reagent (*D. melanogaster*) | γ-tubulin23c-sfGFP | *Tovey and Conduit, 2018* (https://doi.org/10.1016/j.cub.2018.05.044) | | insertion of sfGFP with a 4X GlyGlySer linker at the C-terminus |
| Genetic reagent (*D. melanogaster*) | γ-tubulin23c-eGFP | This paper | | insertion of eGFP with a 4X Ser linker at the C-terminus. Generated by inDroso |
| Genetic reagent (*D. melanogaster*) | sfGFP-Cnn-P1 | This paper | | insertion of sfGFP with a 4X GlyGlySer linker at the start of exon 1a |
| Genetic reagent (*D. melanogaster*) | sfGFP-Cnn-P2 | This paper | | insertion of sfGFP with a 4X GlyGlySer linker at the start of exon 1b |
| Antibody | anti-GFP (mouse monoclonal) | Roche | Cat# 11814460001; RRID:AB_390913 | IF(1:250) |
| Antibody | anti-Cnn (rabbit polyclonal) | *Lucas and Raff, 2007* (https://doi.org/10.1083/jcb.200704081) | | IF(1:1000); gift from Jordan Raff |
| Antibody | anti-Plp (rabbit polyclonal) | *Martinez-Campos et al., 2004* (https://doi.org/10.1083/jcb.200402130) | | IF(1:500); gift from Jordan Raff |
| Antibody | anti-Arl1 (chicken polyclonal) | *Torres et al., 2014* (doi: 10.1242/jcs.122028) | | IF(1:200); gift from Sean Munroe |

*Continued on next page*

*Continued*

| Reagent type (species) or resource | Designation | Source or reference | Identifiers | Additional information |
|---|---|---|---|---|
| Antibody | anti-GM130 (rabbit polyclonal) | Abcam | Cat# ab30637; RRID:AB_732675 | IF(1:300) |
| Antibody | Alexa-647 conjugated Hrp (goat polyclonal) | Jackson | Jackson ImmunoResearch 123-605-021; RRID:AB_2338967 | IF(1:500) |
| Antibody | ATTO-488 GFP-booster (Camelid single domain 'nanobody') | Chromotek | gb2AF488-10; RRID:AB_2827573 | IF(1:200) |
| Antibody | anti-Mouse IgG H and L Alexa Fluor 488 (goat polyclonal) | Abcam | Cat# ab150117, RRID:AB_2688012 | IF(1:500) |
| Antibody | anti-Rabbit IgG H and L Alexa Fluor 568 (goat polyclonal) | Thermo Fisher Scientific | Cat# A-11036, RRID:AB_10563566 | IF(1:500) |
| Recombinant DNA reagent | pCDF3 (plasmid) | Addgene | RRID:Addgene_49410 | pCFD3-dU6:3gRNA; gift from Simon Bullock |
| Recombinant DNA reagent | pBluescript SK+ (plasmid) | Simon Bullock lab | | Standard cloning vector; gift from Simon Bullock |
| Recombinant DNA reagent | pBS-KS-attB1-2-PT-SA-SD-2-sfGFP-FIAsh-StrepII-TEV-3x-FLAG (plasmid) | *Drosophila* Genomics Resource Center | Stock number:1314 | Vector used to amplify sf GFP sequence |
| Sequence-based reagent | CnnNt_P1T2_f | This paper | Guide oligo | GTCGTGTTTAGACTGGTCCATGGG To generate Bbs1 cleavable fragment to clone into Bbs1 cut pCFD3 for Cnn promoter one tagging. Guide sequence underlined. |
| Sequence-based reagent | CnnNt_P1T2_b | This paper | Guide oligo | AAACCCCATGGACCAGTCTAAACA; To generate Bbs1 cleavable fragment to clone into Bbs1 cut pCFD3 for Cnn promoter one tagging. Guide sequence underlined. |
| Sequence-based reagent | CnnNt_P2T2_f | This paper | Guide oligo | GTCGTTAAATGAAACATAGAATA; To generate Bbs1 cleavable fragment to clone into Bbs1 cut pCFD3 for Cnn promoter two tagging. Guide sequence underlined. |
| Sequence-based reagent | CnnNt_P2T2_b | This paper | Guide oligo | AAACTATTCTATGTTTCATTTAA; To generate Bbs1 cleavable fragment to clone into Bbs1 cut pCFD3 for Cnn promoter two tagging. Guide sequence underlined. |
| Sequence-based reagent | cnn_UHA_P1_GFP_f | This paper | PCR primers | AAAGTTAACTATTTGA GGACCTCCCATGGTGT CCAAGGGCGAGGAG; used to amplify sfGFP with overhangs for HiFi cloning into pBS containing homology arms for for Cnn promoter 1 |

*Continued on next page*

Continued

| Reagent type (species) or resource | Designation | Source or reference | Identifiers | Additional information |
|---|---|---|---|---|
| Sequence-based reagent | cnn_DHA_P1_GFP_b | This paper | PCR primers | CCCGCAAAACCTGTTTAGACTGGTCGGATCCGCCGCTACCTCCGCTTCCACCGGAACCTCCCTTGTACAGCTCATCCATGCC; used to amplify sfGFP with overhangs for HiFi cloning into pBS containing homology arms for for Cnn promoter 1. Also contains sequence for 4X GlyGlySer linker |
| Sequence-based reagent | cnn_UHA_P2_GFP_f | This paper | PCR primers | GCAAATGTTAAATGAAAGATACAATATGGTGTCCAAGGGCGAGGAG; used to amplify sfGFP with overhangs for HiFi cloning into pBS containing homology arms for for Cnn promoter 2 |
| Sequence-based reagent | cnn_DHA_P2_GFP_b | This paper | PCR primers | GTGAGGTAGATCGAAAGATACCCGCGGATCCGCCGCTACCTCCGCTTCCACCGGAACCTCCCTTGTACAGCTCATCCATGCC; used to amplify sfGFP with overhangs for HiFi cloning into pBS containing homology arms for for Cnn promoter 2. Also contains sequence for 4X GlyGlySer linker |
| Sequence-based reagent | cnn_UHA_P1_f | This paper | PCR primers | GAAAGCTAAGCAGATTCTTCAGC; used to amplify upstream homology arm for promoter 1 |
| Sequence-based reagent | cnn_UHA_P1_b | This paper | PCR primers | GGGAGGTCCTCAAATAGTTAAC; used to amplify upstream homology arm for promoter 1 |
| Sequence-based reagent | cnn_DHA_P1_f | This paper | PCR primers | GACCAGTCTAAACAGGTTTTGC; used to amplify downstream homology arm for promoter 1 |
| Sequence-based reagent | cnn_DHA_P1_b | This paper | PCR primers | GCTCTCTGCACGTCCAAATAAAC; used to amplify downstream homology arm for promoter 1 |
| Sequence-based reagent | cnn_UHA_P2_f | This paper | PCR primers | CACTCGCTATAGTCTCGATCC; used to amplify upstream homology arm for promoter 2 |
| Sequence-based reagent | cnn_UHA_P2_b | This paper | PCR primers | ATTGTATCTTTCATTTAACATTTGCGCTC; used to amplify upstream homology arm for promoter 2 |

Continued

| Reagent type (species) or resource | Designation | Source or reference | Identifiers | Additional information |
|---|---|---|---|---|
| Sequence-based reagent | cnn_DHA_P2_f | This paper | PCR primers | GCGGGTATCTT TCGATCTACC; used to amplify downstream homology arm for promoter 2 |
| Sequence-based reagent | cnn_DHA_P2_b | This paper | PCR primers | TCCCGTGGAACA GGAACCAC; used to amplify downstream homology arm for promoter 2 |
| Sequence-based reagent | cnn_DHA_P1_pBS_f | This paper | PCR primers | CGGTTTATTTGGACG TGCAGAGAGCGGGG GATCCACTAGTTCTAGAG; used to amplify pBS backbone for promoter 1. Contains overhangs for HiFi cloning. |
| Sequence-based reagent | cnn_UHA_P1_pBS_b | This paper | PCR primers | AAGCTGAAGAATCT GCTTAGCTTTCGG GCTGCAGGAATTCGATATC; used to amplify pBS backbone for promoter 1. Contains overhangs for HiFi cloning. |
| Sequence-based reagent | cnn_DHA_P2_pBS_f | This paper | PCR primers | TTGCTGTGGTTCCTGT TCCACGGGAGGGGGA TCCACTAGTTCTAGAG; used to amplify pBS backbone for promoter 2. Contains overhangs for HiFi cloning. |
| Sequence-based reagent | cnn_UHA_P2_pBS_b | This paper | PCR primers | AAGTGGATCGAGAC TATAGCGAGTGGG GCTGCAGGAATT CGATATC; used to amplify pBS backbone for promoter 2. Contains overhangs for HiFi cloning. |
| Other | Live imaging solution | Thermo Fisher Scientific | A14291DJ | Larvae imaging |

## Contact for reagent and resource sharing

Further information and requests for resources and reagents should be directed to and will be fulfilled by the Lead Contact, Paul Conduit (paul.conduit@ijm.fr).

## Experimental model and subject details

All fly strains were maintained at 18 or 25°C on Iberian fly food made from dry active yeast, agar, and organic pasta flour, supplemented with nipagin, propionic acid, pen/strep and food colouring.

### Drosophila melanogaster stocks

The following fluorescent alleles were used in this study: γ-tubulin23C-sfGFP (*Tovey et al., 2018*), γ-tubulin23c-eGFP (this study), sfGFP-Cnn-P1 (this study), sfGFP-Cnn-P2 (this study), UAS-mCD8-mRFP (BL 27392), UAS-EB1-GFP (BL 35512), UAS-ManII-mCherry (Yuh-Nung Jan), and UAS-ManII-GFP (BL 65248). The following Gal4 lines were used in this study: ppk-Gal4 Chr II (B32078) and Chr III (BL 32079) and 221-Gal4 (BL 26259). The following mutant alleles were used in this study: *plp*[5] (BL

9567), *plp*<sup>s2172</sup> (BL 12089), *cnn*<sup>f04547</sup> (Exelixis at HMS), ppkCD4-tdGFP (BL 35842), KAP RNAi (VDRC 45400 GD), γ-tubulin37C RNAi (BL32513), Klp64D RNAi (VDRC 45373) UAS-Dicer 2 (BL 24646).

For examining the endogenous localisation of γ-tubulin23C we used flies expressing γ-tubulin23C-sfGFP and γ-tubulin23C-eGFP i.e. 2 copies of γ-tubulin23C-GFP, either alone or in combination with one copy of UAS-mCD8-RFP expressed under the control of one copy of either 221-Gal4 or ppk-Gal4. For examining the localisation of Cnn, we used flies expressing two copies of either sfGFP-Cnn-P1 or sfGFP-Cnn-P2 either alone or in combination with one copy of UAS-mCD8-RFP expressed under the control of one copy of either 221-Gal4 or ppk-Gal4. For examining the localisation of Plp in relation to medial Golgi, we used flies with one copy of UAS-ManII-GFP expressed under the control of one copy of either 221-Gal4 or ppk-Gal4. For examining the localisation of γ-tubulin23C in the absence of Cnn or Plp, we used flies expressing two copies of γ-tubulin23C-(sf/e) GFP (as above) in a *cnn*<sup>f04547</sup>/*cnn*<sup>f04547</sup> or *plp*<sup>5</sup>/*plp*<sup>s2172</sup> mutant background. For examining microtubule dynamics in relation to the Golgi we used flies with one copy of UAS-EB1-GFP and one copy of UAS-ManII-mCherry, both expressed under the control of one copy of 221-Gal4. For examining microtubule dynamics in control neurons we used flies with one copy of UAS-EB1-GFP and one copy of UAS-γ-tubulin37C RNAi, with or without one copy of UAS-Dicer-2, expressed under the control of one copy of 221-Gal4. For examining microtubule dynamics in Kap3 RNAi neurons we used flies with one copy of UAS-EB1-GFP, one copy of UAS-Dicer-2, and one copy of UAS-Kap3 RNAi expressed under the control of one copy of 221-Gal4.

## Method details
### DNA cloning
5-alpha Competent *E. coli* (High Efficiency) (NEB) cells were used for bacterial transformations, DNA fragments were purified using QIAquick Gel Extraction Kits (Qiagen), plasmid purification was performed using QIAprep Spin Miniprep Kits (Qiagen). Phusion High-Fidelity PCR Master Mix with HF Buffer (ThermoFisher Scientific) was used for PCRs.

## Generating endogenously-tagged fly lines
All endogenously-tagged lines were made using CRISPR combined with homologous recombination, by combining the presence of a homology-repair vector containing the desired insert with the appropriate guide RNAs and Cas9. The *γ-tubulin23C-eGFP* allele was generated by inDroso by initially inserting an SSSS-eGFP-3'UTR-LoxP-3xP3-dsRED-Lox P cassette before the selection markers were excised. The multi-serine insert acts as a flexible linker between γ-tubulin23C and eGFP. The following guide RNA sequences were used to cut either side of the 3'UTR: AGTCGATC|TGTGAC-CAGCGC and TTATGGTT|AATGTCGACTTG. The sfGFP-Cnn-P1 (insertion of sfGFP at the start of exon 1a) and sfGFP-Cnn-P2 (insertion of sfGFP at the start of exon 1b) alleles were generated within the lab following a similar approach to that used previously (**Tovey et al., 2018**). For GFP-Cnn-P1, flies expressing a single guide RNA containing the target sequence TGTTTAGACTGGTCCATGGG were crossed to nos-Cas9 expressing females and the resulting embryos were injected by the Department of Genetics Fly Facility, Cambridge, UK, with a pBluescript plasmid containing sfGFP and linker sequence (4X GlyGlySer) flanked on either side by 1.5 kb of DNA homologous to the *cnn* genomic locus surrounding the 5' end of the appropriate coding region. The homology vectors were made by HiFi assembly (NEB) of PCR fragments generated from genomic DNA prepared from nos-Cas9 flies (using MicroLYSIS, Microzone) and a vector containing the sfGFP tag (DGRC, 1314). For GFP-Cnn-P2 flies, both the guide RNA containing the target sequence GTTAAATGAAACATAGAA TA and the homology vector were injected into nos-Cas9 embryos (BL54591) by Rainbow Transgenic Flies, Inc Camarillo, CA 93012, USA. F1 and F2 males were screened by PCR using the following primers: for sfGFP-Cnn-P1: forward primer: AAAGTTAACTATTTGAGGACCTCCCATGGTG TCCAAGGGCGAGGAG; reverse primer: CCCGCAAAACCTGTTTAGACTGGTCGGATCCGCCGC TACCTCCGCTTCCACCGGAACCTCCCTTGTACAGCTCATCCATGCC; for sfGFP-Cnn-P2: forward primer: GCAAATGTTAAATGAAAGATACAATATGGTGTCCAAGGGCGAGGAG; reverse primer: G TGAGGTAGATCGAAAGATACCCGCGGATCCGCCGCTACCTCCGCTTCCACCGGAACCTCCCTTG TACAGCTCATCCATGCC.

## Antibodies

The following primary antibodies were used: anti-GFP mouse monoclonal at 1:250 (Roche, 11814460001), anti-Cnn Rabbit polyclonal raised against first 660aa of Cnn-P1 (which includes amino acids 35–632 of Cnn-P2) at 1:1000 (*Lucas and Raff, 2007*), anti-Plp rabbit polyclonal at 1:500 (*Martinez-Campos et al., 2004*), Alexa-647 conjugated HRP polyclonal at 1:500 (Jackson), anti-Arl1 chicken polyclonal at 1:200 (gift from Sean Munro *Torres et al., 2014*), and anti-GM130 rabbit polyclonal at 1:300 (Abcam). The following secondary antibodies were used: ATTO-488 GFP-booster at 1:200 (Chromotek), Alexa-488 anti-Mouse at 1:500 (Abcam), Alexa-547 anti-Rabbit at 1:500 (ThermoFisher).

## Immunostaining

Larvae were dissected as described previously (*Broadie and Bate, 1993*). The fillet preparations were fixed in freshly prepared 4% formaldehyde for 20 min at room temperature and were then washed four times for 10 min in PBST (PBS + 0.1% TritonX-100). The preparations were then blocked in PBST + 5% BSA for 1 hr at room temperature and incubated with the appropriate primary antibodies diluted in PBST overnight at 4°C. After washing in PBST for ~8 hr, changing washes every 30–45 min, samples were incubated in secondary antibodies diluted in PBST overnight at 4°C. The fillet preparations were then washed for ~8 hr, changing washes every 30–45 min in PBST before mounting in Moviol. They were stored at −20°C and imaged within a week. Neurons within segments A2 to A6 were imaged; we observed no difference in the staining patterns between these segments.

## Fixed and live cell imaging

Imaging of all samples except for those expressing EB1-GFP in neurons or GFP-Cnn-P1 in embryos was carried out on an Olympus FV3000 scanning inverted confocal system run by FV-OSR software using a 60 × 1.4 NA silicon immersion lens (UPLSAPO60xSilcon). For live samples, wandering L3 larvae were placed in a drop of glycerol and flattened between a slide and a 22 × 22 mm coverslip, held in place by double-sided sticky tape, and imaged immediately. Imaging of EB1-GFP within da neurons and GFP-Cnn-P1 within embryos was performed on a Leica DM IL LED inverted microscope controlled by μManager software and coupled to a RetigaR1 monochrome camera (QImaging) and a CoolLED pE-300 Ultra light source using a 63X 1.3 NA oil objective (Leica 11506384). For EB1-GFP imaging, larvae were dissected in live imaging solution (ThermoFisher) to remove the majority of their tissues and mounted in Schneider's medium supplemented with FBS and pen/strep between a slide and 22 × 22 mm coverslip held in place with tape and imaged immediately. When using the CherryTemp, the larvae were held between the CherryTemp chip and a 22 × 22 mm coverslip. The CherryTemp software was used to rapidly change the temperature of the liquid within the flow chamber from 20°C to 5°C after an initial 45 s of imaging. The temperature was then rapidly changed back to 20°C after a further 180 s and the neurons were imaged for a further 135 s. For EB1-GFP imaging, single Z-plane images were acquired every 5 s; for EB1-GFP/ManII-mCherry dual imaging single Z-plane images were acquired every 3 s. All images were processed using Fiji (ImageJ). EB1 comets were tracked using the Manual Tracking plugin in Fiji.

## Quantification and statistical analysis

Statistical analysis and graph production were performed using GraphPad Prism and SciStat.com.

To determine punctate versus diffuse localisation of γ-tubulin-GFP at branchpoints of class I da neurons, we visually categorised each branchpoint into one or other group depending on whether the puncta were clear and obvious or not. If they were clear and obvious, then they were categorised as puncta. If puncta and diffuse patches did co-exist we defined them as puncta. To quantify the number of Golgi outposts and γ-tubulin-GFP puncta per 100 μm dendrite we measured the total length of dendrites in ImageJ using the segmented line tool and marked and counted the number of Golgi outposts and γ-tubulin-GFP puncta. We then calculated their number per 100 μm dendrite. We did this across multiple images of different neurons. We analysed 14 class I neurons with a total dendritic length of 8525 μm and a total number of 183 branchpoints. We analysed 10 proximal and seven distal images of class IV da neurons with a total dendritic length of 3619 μm and 4489 μm, respectively, and a total number of 47 and 179 branchpoints, respectively.

For EB1-comet analysis, we excluded comets that were present within the first timepoint (and thus may not represent newly growing microtubules). The angle of initial comet growth was measured using the angle tool within ImageJ, by drawing lines from the axon entry site to the comet origin and then along the initial linear path of the comet. For the frequency distribution of comet angles in *Figure 6B*, the negative angles were made positive, so as not to distinguish between comets growing either side of the axon entry site, creating a distribution between 0° and 180°. A total of 163 comets from 59 Golgi stacks from 7 cells were analysed.

For the vector analysis in *Figure 6C,D* we used both positive and negative angles (between −180° and 180°). Initially, a vector for each comet was generated with a length of 1 by calculating cosine(angle) (Y value) and sin(angle) (X value). Taking in turn each Golgi stack that generated more than one comet, the vectors of each comet were added together to generate a resultant vector. These resultant vectors were then normalised by dividing their X and Y values by the number of comets originating from the Golgi stack, such that the maximum length of the resultant vector would be 1, irrespective of comet number. These normalised resultant vectors were plotted as circles on a scatter plot, with the size of each circle reflecting the number of comets originating from that particular Golgi stack. 44 Golgi stacks were analysed from seven neurons. We generated random sets of data by replacing each comet angle with a randomly generated number between −180 and 180 to show what results could be expected if the angles of comets originating from Golgi stacks were independent of each other and of the position of the axon (thus each random set of data was generated using 44 hypothetical Golgi stacks). We generated three random sets of data, the resultant vector length distributions of which are included in *Figure 6D*.

One-way Chi-squared tests were used to determine whether frequency distributions were significantly different from the distribution expected by chance i.e. whether they were different from random comet angles; a binomial probability calculator was used to determine the chance of observing the skewed distribution between upper (towards the axon) and lower (away from the axon) quadrants in the resultant vector scatter plot in *Figure 6C*.

To assess whether comets that originated within the soma approached and entered axons or dendrites, we defined a region (~0.5 μm wide) across the entrance to the axon or dendrite; any comets that entered this region were scored as comets that had approached; any comets that crossed this region were scored as comets that entered. For control neurons, we analysed a total of 666 comets (across 13 movies) that had originated during the movies within the soma (average of 10.0 comets per minute); for Kap3 RNAi we analysed a total of 1058 comets (across nine movies) (average of 20.0 comets per minute). Due to the labour-intensive nature of manual tracking we did not track all comets within the soma of Klp64D RNAi neurons; we instead focussed only on comets that approached dendrites (a total of 80 comets from 10 movies). To assess the polarity of dynamic microtubules within the proximal dendritic regions, we included both comets that originated within the proximal dendrite and those that grew into the dendrite from the soma. For control neurons, we analysed a total of 252 comets from 13 cells; for Kap3 RNAi we analysed a total of 338 comets from 9 cells. The overall proportion of comets that either turned, approached, entered or had anterograde polarity was plotted in *Figure 7C–F* along with the 95% confidence interval for each set of data. Sets of data were compared using one-way Chi-squared tests.

## Acknowledgements

This work was supported by a Wellcome Trust and Royal Society Sir Henry Dale Fellowship (105653/Z/14/Z) and an Isaac Newton Trust Research grant (18.23(p)) awarded to PTC, and an Association pour la Recherche sur le Cancer grant (PJA 20181208148) awarded to AG. We thank Sam Comb for help with cloning for the endogenously-tagged fly lines. We thank other members of the Conduit lab for their invaluable input and critical reading of the manuscript. We thank Matthias Landgraf, Melissa Rolls, Adrian Moore, Jill Willdonger for fruitful discussion during the course of the project. We thank Sean Munro and Alison Gillingham for advice on the Golgi and providing us with the Arl1 antibody. We thank Jordan Raff for supplying various antibodies. We thank Darren Williams, Guy Tear, Bing Ye and Yuh-Nung Jan for fly lines that were used in this study. The work benefited from use of the Imaging Facility, Department of Zoology, supported by Matt Wayland and a Sir Isaac Newton Trust Research Grant (18.07ii(c)).

## Additional information

### Funding

| Funder | Grant reference number | Author |
|---|---|---|
| Wellcome | 105653/Z/14/Z | Amrita Mukherjee<br>Paul S Brooks<br>Paul T Conduit |
| Isaac Newton Trust | 18.23(p) | Amrita Mukherjee<br>Paul T Conduit |
| Association pour la Recherche sur le Cancer | PJA 20181208148 | Fred Bernard<br>Antoine Guichet |

The funders had no role in study design, data collection and interpretation, or the decision to submit the work for publication.

### Author contributions

Amrita Mukherjee, Paul T Conduit, Conceptualization, Resources, Data curation, Formal analysis, Supervision, Funding acquisition, Validation, Investigation, Visualization, Methodology, Writing - original draft, Project administration, Writing - review and editing; Paul S Brooks, Antoine Guichet, Resources, Data curation, Formal analysis, Investigation, Methodology; Fred Bernard, Resources

### Author ORCIDs

Fred Bernard (iD) http://orcid.org/0000-0002-7919-253X
Antoine Guichet (iD) http://orcid.org/0000-0001-7216-1944
Paul T Conduit (iD) https://orcid.org/0000-0002-7822-1191

### Decision letter and Author response

Decision letter https://doi.org/10.7554/eLife.58943.sa1
Author response https://doi.org/10.7554/eLife.58943.sa2

## Additional files

### Supplementary files

• Transparent reporting form

### Data availability

All data generated or analysed during this study are included in the manuscript and supporting files. Source data files have been provided for Figure 6 and 7.

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
