## [Decision Letter]

**Acceptance summary:**

This study describes microtubule nucleation at the somatic Golgi in the *Drosophila* neurons, which explains how microtubule polarity may be regulated in dendrites and axons. This is an important study that provides new insights into a long-standing observation of microtubule polarity in neurons. During the revision, the authors addressed reviewer concerns very well, also clearly stating what will be required to further test the functionality of microtubule nucleation at the somatic Golgi (and we fully recognize that localization-specific function of a given protein is difficult to prove for any given molecules). Also, circumstantial evidence provided in the paper is sufficiently strong to propose the function of Golgi-localized microtubule nucleation.

**Decision letter after peer review:**

[Editors’ note: the authors submitted for reconsideration following the decision after peer review. What follows is the decision letter after the first round of review.]

Thank you for submitting your work entitled "Asymmetric nucleation and guidance maintain microtubule polarity within neurons" for consideration by *eLife*. Your article has been reviewed by two peer reviewers, and the evaluation has been overseen by a Reviewing Editor and a Senior Editor. The reviewers have opted to remain anonymous.

Our decision has been reached after consultation between the reviewers. Based on these discussions and the individual reviews below, we regret to inform you that your work will not be considered further for publication in *eLife*.

The reviewers appreciated the importance of the questions being addressed in the manuscript. However, both raised concerns on lack of functional analysis on the major part of the manuscript. As you can see in individual comments, the reviewers felt that addressing these concerns will be beyond the scope of straightforward revision permitted at *eLife*. However, we would be happy to reconsider your revised manuscript as a new submission if the major concerns are thoroughly addressed.

Reviewer #1:

This study addresses the important problem of microtubule organization in *Drosophila* dendrites using class I and class IV da neurons. Two main questions are addressed in the paper: (i) the nature of microtubule nucleation sites; (ii) polarity of growing microtubules. I would like to discuss these two questions separately.

1) Microtubule nucleation. The authors demonstrated that the majority of the nucleation events are happening at the γ-tubulin-containing Golgi outposts in class IV neurons and that localization of γ-TuRC to Golgi is independent of Centrosomin or Pericentrin-like protein. The paper contains good data demonstrating the nucleation on Golgi outposts, but I am not convinced that they clearly demonstrated that γ-TuRC localization to Golgi is independent of Cnn or Plp. Their conclusion is exclusively based on localization data, but it is not clear that the sensitivity of their colocalization techniques is sufficient for this conclusion. I would feel more confident if the microscopy results are supplemented with Cnn and Plp knock-down experiments in da neurons.

On a technical note here, the temperature shift experiment that they use to precisely localize microtubule nucleation sites is much less perfect than the authors believe. Most, if not all cells contain a sizable fraction of cold-stable microtubules (these microtubules are often acetylated). These microtubules by definition can survive the cold treatment and serve as nucleation sites when the temperature is shifted back, thus obscuring the correct location of nucleation sites in untreated neurons.

2) The polarity of the microtubule growth.

This part of the work extends the classical observation of Melissa Rolls and colleagues (Mattie et al., 2010) demonstrating that Kinesin-2 through EB1 and APC moves plus ends of growing microtubules toward plus ends of existing microtubules, thus promoting uniform polarity of microtubules.

The difference between this paper and the original paper and Mattie et al. is that here the authors are mostly looking at the growth of microtubules originated in the cell body and entering the axons while Mattie et al. studied uniform (minus-end-out) microtubule polarization in dendrites. Unfortunately, neither of these two papers addresses the key question of how the polarity in the dendrites and in the axons is first established, but only the question of how it is maintained. Whether the advances of this part of the manuscript justify its publication is not quite clear. What would make this part of the paper stronger is the imaging of EB1 and microtubules at the same time (the authors can label microtubules by tau-GFP or Jupiter-GFP) to show that EB1 on plus ends of growing microtubules indeed tracks pre-existing microtubules. In addition, the hypothesis that APC serves as a link between the two should be tested and discussed.

Furthermore, the microtubules in dendrites of da neurons span >100 μm distance, and there must be some de novo microtubule nucleation sites within the dendrites (outside of the soma). In addition to the somatic Golgi model, the authors should discuss these dendritic nucleation sites and their roles in maintaining the microtubule polarity in neurons.

Reviewer #2:

The authors describe some nice tools to look at endogenous microtubule regulators including γ-tubulin, Cnn and Plp. The first 6 figures are focused on trying to pin down sites of microtubule nucleation in *Drosophila* neurons using these tools. However, without any data on how the localization they see relates to nucleation (only Figure 6 tries to do this, and it needs substantial additional controls), it is difficult to know what the different spots, patches etc that they see mean. In Figure 7, behavior of growing microtubules in the cell body is analyzed, and here functional data is presented for the Kinesin-2 subunit Kap3. Overall it seems like two rather unrelated stories with functional data only for the second one. This is reflected in the Abstract, which is focused on the function of Kinesin-2, but not the figures, as only Figure 7 relates to this part of the story.

Major points:

1) The authors describe localization of γ-tubulin-GFP to puncta, diffuse spots, branch points and dendritic bubbles in Figures 1-5. The functional significance of these localizations is not clear and largely unexplored in the manuscript.

2) The first functional data is presented in Figure 6 with cold treatment of larvae to depolymerize microtubules. However, the key control of showing the treatment actually does depolymerize microtubules is missing.

3) The data in Figure showing that Kinesin-2 may help prevent microtubules growing into dendrites is intriguing but based on quite low numbers of events and only one RNAi condition. This part of the manuscript is not very closely related to the rest of the work.

Points on individual figures:

Figure 1: The authors distinguish γ-tub-GFP puncta and diffuse patches. How were these distinguished? Is it known whether one is more functionally relevant than the other? In the example images in B, which are puncta, and which are diffuse? I did not see a description of how the different types of signals were quantitated although numbers are mentioned in the text. In the legend it is noted that images in B and C are from living animals. It is a bit confusing whether A is also live. It looks like the signal in fixed neurons is more punctate than in live ones. Is this an accurate impression? If so, it would be good to mention in the text.

Figure 2: It looks like there is a movement artifact at the right side of the images in A and A' (double spots), so it might be good to choose a different example.

The authors conclude: "Collectively, our data show that Golgi outposts within class IV neuron branchpoints close to the soma frequently associate with γ-TuRCs," However, no data on γ-TuRCs was presented – are the authors assuming that the spots where they see γ-tubulin-GFP are γ-TuRCs? This claim should be supported either by looking at other γ-TuRC subunits or with functional analysis of nucleation.

While quantitation of Golgi and γ-tub-GFP is mentioned in the text, it is not clear how the quantitation was performed, and I could not find a description of it in the Materials and methods. It would be helpful to include graphs of the quantitation.

Figure 3. The authors stain γ-tubulin-GFP larvae with HRP and antibodies to GM130 and observed some intriguing patterns in the cell body. They conclude "this suggests that γ-tubulin-GFP localises on the rims of the cis-Golgi stack." It would be helpful to support this conclusion with other markers. Is the association with cis-Golgi higher than that with a trans-Golgi marker? Without labeling other parts of the Golgi, or surrounding structures like ER exit sites, it is hard to know whether the spots around the Golgi are most closely associated with cis-Golgi. It is particularly important to pin this down as Plp, which was previously shown to link nucleation sites to the Golgi, seems not to have a role here (Figure 5).

Figure 6. The authors examine the site of new EB1 comet formation in the cell body of class I neurons. How do they know which comets are the result of catastrophe rescue and which are nucleation? If they want to link comet formation to nucleation, it is important to show that it is altered in γ-tubulin mutants.

Some comets were determined to originate from the Golgi. The marker used for the Golgi was ManII, which earlier was described as not localizing cleanly to the Golgi in Class I neurons, which were used for this assay. Overall it is very difficult to know whether the new comets counted in this figure represent nucleation and whether they derive from the Golgi. A bias towards the axon would be very easy to explain based on catastrophe rescue of microtubules that derive from the dendrites, as these grow into the soma towards the axon.

For the microtubule cold depolymerization it is important to show that stable microtubules are actually depolymerized in neurons. They tend to be very refractory to cold or drug-induced depolymerization and unless stable microtubules really are eliminated by this treatment regrowth could be from remaining microtubules rather than nucleation. An additional confirmation that this assay is a good readout of nucleation would be to perform it in γ-tubulin mutant or knockdown neurons. Their statement "any comets that appear a significant amount of time after warming represent new nucleation events" requires significantly more validation. This section of the text also lacks references, for example on whether cold treatment has previously been shown to depolymerize neuronal microtubules.

Figure 7. The title of this section is "Kinesin-2-dependent plus-end turning and dendritic exclusion maintain microtubule polarity in axons and dendrites." Kinesin-2 also functions more distally in dendrites to control direction of microtubule growth. The section title attributes change in polarity to the cell body rather than this function. Can the data the authors present link dendrite polarity changes to a function in the cell body rather than in the dendrites? Is it because they only analyze the proximal dendrite?

There is no data on microtubule polarity in axons, although the section title states that Kinesin-2 has a role in maintaining axon microtubule polarity. Are there changes in axonal polarity in the Kap3 knockdown?

The major finding here, that comets normally do not grow easily in dendrites is interesting. It would be helpful to have stronger data to link Kap3 steering to keeping microtubules away from the dendrite; the increase in dendrite approaches is not quite significant (c). It would be very helpful to analyze more animals to see whether this will hold up, and the numbers of plus ends analyzed is quite low. The increase in successful entries into the dendrite (D) in the Kap3 RNAi is more convincing, but how does this relate to the model? Similarly, if Kinesin-2 was helping microtubules grow along parallel microtubules, would it not be expected that the successful entries into the axon in D would go down?

In the text the authors mention some additional data on turning within the cell body that is not shown in the figure. It would be good to have this in the figure as it adds additional support to the hypothesis that Kinesin-2 is functioning to control microtubule growth in the cell body.

[Editors’ note: further revisions were suggested prior to acceptance, as described below.]

Thank you for submitting your article "Microtubules originate asymmetrically at the somatic Golgi and are guided via Kinesin2 to maintain polarity in neurons" for consideration by *eLife*. Your article has been reviewed by two peer reviewers, and the evaluation has been overseen by a Reviewing Editor and K VijayRaghavan as the Senior Editor. The reviewers have opted to remain anonymous.

The reviewers have discussed the reviews with one another and the Reviewing Editor has drafted this decision to help you prepare a revised submission.

Summary:

This study describes microtubule nucleation at the somatic Golgi in the *Drosophila* neurons, which explains how microtubule polarity may be regulated in dendrites and axons. This is an important study that provides new insights into a long-standing observation of microtubule polarity in neurons.

This is a resubmission of a previously rejected paper. The authors addressed the concerns raised in the previous submission as much as possible, under covid-19 lockdown. One reviewer remains concerned that the functionality of Golgi-localized g-tubulin is not clearly demonstrated, but we recognize that this is generally a very difficult question to address (localization-specific function of a given protein). And other circumstantial evidence provided in the paper is sufficiently strong to propose the function of Golgi-localized microtubule nucleation.

Given the circumstance and the current editorial policy at *eLife*, we would like to invite to submit a revised version that addresses remaining reviewer concerns textually as much as possible. In the future, when the authors can obtain additional experimental data to support the conclusion, the paper should be amended by those new data.

Reviewer #1:

The authors have done everything they could under the current conditions. In normal times, I would certainly require simultaneous imaging of plus ends and a microtubule marker at the same time, this would make the paper much stronger. However, this type of imaging is not trivial, and it would be unfair to require it now. I think we can accept the manuscript.

Reviewer #2:

This manuscript is a resubmission of a previous version that was rejected. Unfortunately, the key functional experiments that were suggested in the previous comments were not performed because of Covid-related lab closure. While some leeway seems reasonable for manuscripts close to acceptance, asking key data to be overlooked for a rejected manuscript seems a stretch. While the authors were able to increase the n on one experiment and provide some additional staining, as they mention in the rebuttal many of the important experiments have not been done yet. Therefore, the limitations in the original submission remain.

Examples of key issues that have not been changed in the new version:

Much of the data remains descriptive without functional backup. For example, γ-tubulin localization to puncta vs diffuse concentrations is described, but it is not clear whether one type of localization is functional, or both are. Similarly, γ-tubulin is described as localizing to dendrite bubbles, and again this is unconnected to function.

Analysis of microtubule behavior in the cell body of γ-tubulin mutants is important to demonstrate that the comets initiating near Golgi are due to nucleation. In their live experiments with Golgi and EB1 (Figure 6) about half of the cell body looks like it is occupied by the Golgi marker. I did not see any quantitation to show that new comet formation is enriched on Golgi compared to other areas of cytoplasm. It is important to do this in control and γ-tubulin mutant animals.

The temperature shift assay remains difficult to interpret without additional information about what microtubules remain after cooling. There also does not seem to be quantitation in the figures to support conclusions made from this data.

---

## [Author Response]

[Editors’ note: the authors resubmitted a revised version of the paper for consideration. What follows is the authors’ response to the first round of review.]

Below we provide a detailed response to each of the reviewers’ comments, explaining how we have either directly addressed their concerns with new data or with detailed explanations within the text. These textual changes include clearer arguments as to why our data strongly supports our conclusions and also discussions of which future experiments will help further support the conclusions, as per eLife’s current publishing policy. In correspondence with the editors, we had highlighted aspects of the paper that reviewer 1 appeared to have misunderstood. Namely they stated that we had shown microtubule nucleation from Golgi outposts in class IV neurons. However, while we showed that γ-tubulin localises to a few proximal Golgi outposts in class IV neurons, our data showed microtubule nucleation from the somatic Golgi. In addition, experiments on γ-tubulin-GFP localisation in the absence of Cnn and Plp and additional discussion on dendritic microtubule nucleation were requested, however these were already present in the manuscript. In response to this, we were told that *“reviewer #1 acknowledges that there was some misunderstanding on their part, thus the parts concerning this misunderstanding do not have to be addressed during the revision”* – we have therefore excluded these comments from this rebuttal letter.

Reviewer #1:This study addresses the important problem of microtubule organization in Drosophila dendrites using class I and class IV da neurons. Two main questions are addressed in the paper: (i) the nature of microtubule nucleation sites; (ii) polarity of growing microtubules. I would like to discuss these two questions separately.1) Microtubule nucleation. The authors demonstrated that the majority of the nucleation events are happening at the γ-tubulin-containing Golgi outposts in class IV neurons and that localization of γ-TuRC to Golgi is independent of Centrosomin or Pericentrin-like protein. The paper contains good data demonstrating the nucleation on Golgi outposts, but I am not convinced that they clearly demonstrated that γ-TuRC localization to Golgi is independent of Cnn or Plp. Their conclusion is exclusively based on localization data, but it is not clear that the sensitivity of their colocalization techniques is sufficient for this conclusion. I would feel more confident if the microscopy results are supplemented with Cnn and Plp knock-down experiments in da neurons.On a technical note here, the temperature shift experiment that they use to precisely localize microtubule nucleation sites is much less perfect than the authors believe. Most, if not all cells contain a sizable fraction of cold-stable microtubules (these microtubules are often acetylated). These microtubules by definition can survive the cold treatment and serve as nucleation sites when the temperature is shifted back, thus obscuring the correct location of nucleation sites in untreated neurons.

This point surrounds the main concern that we have not yet proven microtubules are nucleated from the somatic Golgi, which was also a concern of reviewer 2. So that we do not repeat ourselves, we detail our response to this main concern here.

Firstly, we have new experimental data that we believe supports our claim that the somatic Golgi nucleates microtubules. By staining different Golgi compartments, we have now convincingly shown that γ-tubulin localises asymmetrically to the cis-Golgi compartment of each Golgi stack (new Figure 3D). This fits with our observation that microtubules grow out asymmetrically from the somatic Golgi, further supporting our conclusion that the somatic Golgi nucleates microtubules asymmetrically. This is very similar to the situation in some mammalian cells, such as fibroblasts, where γ-tubulin localises asymmetrically to the cis-Golgi compartment within the soma and where microtubules are nucleated asymmetrically from the Golgi (see Rios, 2014 for a Review of the literature). We hope the reviewers agree that it would be surprising that the mammalian somatic Golgi but not the *Drosophila* da neuron somatic Golgi would nucleate microtubules when both recruit γ-tubulin in a similar fashion. Because the comparison to mammalian cells supports our conclusion, we have explained this clearly in the Discussion of the new manuscript.

Before the Covid-19 crisis, we had planned to directly address the reviewers’ concern by assessing the degree of microtubule depolymerisation during cold treatment within the cell bodies of neurons expressing Jupiter-mCherry, a microtubule binding protein. Clemens Cabernard kindly sent us a UAS-Jupiter-mCherry line, but unfortunately this did not arrive until 23^rd^ March, after our University had closed.

In the absence of this new experimental data, we have therefore modified the text in the paper to be more cautious in interpreting our results, but we have also expanded our arguments to make it clearer that our current data, taken collectively, still strongly supports the conclusion that microtubules are nucleated from the somatic Golgi. While we still agree with the reviewers that stable microtubules may not have been depolymerised in our cooling-warming microtubule nucleation assay, one would expect that most highly dynamic microtubules (including those that were recently nucleated) would have been depolymerised by cold treatment. This prediction fits with the observation that EB1-GFP comets disappear immediately on cooling, which indicates the loss of their GTP-cap, which is required to prevent microtubule depolymerisation of a dynamic microtubule. Our results that several comets emerge from the Golgi stacks immediately after warming, and that these comets do not emerge from the same place that comets had stopped on cooling, is indicative that at least a fraction of microtubules are nucleated from the Golgi. It is surely unlikely that these comets could all represent re-growing stable microtubules that just happened to regrow from sites overlapping the Golgi.

In our opinion, the best evidence we have is that comets emerge from the Golgi over 50s after warming, and some of these comets do not emerge from the same Golgi that had generated a comet immediately after warming. The direction of all the late comets does not suggest that they were simply re-growing microtubules that had polymerised on warming, then depolymerised, and then had grown again from sites overlapping the Golgi. We realise that we had not made this clear in the first submission and so we have now done so in the new manuscript.

We also predict that many cold-stable acetylated microtubules would remain stable rather than become dynamic after cooling and re-warming. If true, they can essentially be ignored in this analysis. There is also an argument for not wanting to depolymerise all microtubules, as depolymerising all microtubules may result in an unusually high concentration of free tubulin that could, in theory, lead to nucleation from sites that do not normally nucleate microtubules under physiological conditions (Pickett-Heaps et al., 1982). Thus, to the best of our ability, we have used the cooling-warming microtubule nucleation experiment to complement our other data (asymmetric γ-tubulin localisation, asymmetric emergence of EB1-GFP comets from the Golgi) that strongly suggests microtubules are nucleated from the somatic Golgi. We have now included the experiment under its own sub-heading: “A Temperature-based microtubule nucleation assay suggests that microtubules are nucleated from the somatic Golgi in class I da neurons” to allow us to include our more expansive arguments.

Most importantly, our results from the cooling-warming assay should not be taken in isolation. We have also shown that γ-tubulin localises strongly to the somatic Golgi, that this localisation is asymmetric (new Figure 3D), and that EB1-GFP comets repeatedly grow out from Golgi stacks in a similar direction to each other (Figure 6). This is similar to what occurs in mammalian cells where it is well established that microtubules are nucleated from the somatic Golgi. Several papers investigating microtubule nucleation in da neurons have previously considered the emergence of comets from a single site, especially the repeated emergence of comets in a similar direction from the same site, as representative of nucleation (Zhou et al., 2014; Ori-McKenney 2014; Nguyen et al., 2014; Yalgin et al., 2015; Weiner et al., 2020). These studies did not robustly examine the localisation of endogenous γ-tubulin or perform cold-treatment experiments, so we have actually gone a step further in trying to establish genuine nucleation sites. Thus, while we have been unable to perform all requested experiments and while we cannot rule out other less likely possibilities as to why we see comets emerging from the Golgi, we can provide strong evidence to support our conclusions.

Nevertheless, we still appreciate the reviewers’ concern and have therefore made the following changes in the new manuscript:

– We have changed our title and Abstract to avoid using the word “nucleation” and to add elements of caution to our conclusions.

– We have modified the summary paragraph at the end of the Introduction to tone down our conclusion that the somatic Golgi nucleates microtubules: “…Tracking of EB1-GFP comets within the soma suggests that microtubules are nucleated asymmetrically from the somatic Golgi…”

– We have changed the Results title from “The somatic Golgi within da neurons nucleates microtubules asymmetrically” to “Growing microtubule originate asymmetrically from the somatic Golgi”.

– In the Results section we have more tentatively made the point that repeated growth from the same location in a similar direction is indicative of microtubule nucleation: “While not definitive, the similarity in the direction of comets emerging repeatedly from the same Golgi stack is indicative of consecutive microtubule nucleation events”.

– In the Results section we have modified “Cooling causes depolymerisation of dynamic microtubules and warming causes their regrowth from their nucleation site” to “Cooling typically causes depolymerisation of dynamic microtubules and warming causes their regrowth”.

– We have created a new section for the cooling-warming microtubule nucleation experiment to allow a more detailed analysis and discussion of the observations. This section is entitled: “A Temperature-based microtubule nucleation assay suggests that microtubules are nucleated from the somatic Golgi in class I da neurons”. This now includes discussing the possibility that a fraction of comets emerging after cooling-warming could have been from catastrophe-rescue of cold-stable microtubules. It also includes an expanded argument as to why the late emerging comets provide good evidence for nucleation from the Golgi.

– We have also substantially modified and expanded the discussion to make our arguments clearer, to discuss potential caveats with our experiments, and to discuss possible future experiments.

In addition, and with the reviewers concerns firmly in our minds, we want to make it clear that we plan to perform the additional experiments proposed in our originally appeal (testing microtubule depolymerisation on cooling and testing the effect of depleting γ-TuRCs on microtubule growth from the Golgi) and endeavour to publish the results in a follow up paper that will be linked to this current paper, as recommended in *eLife*’s new publishing guidelines.

2) The polarity of the microtubule growth.This part of the work extends the classical observation of Melissa Rolls and colleagues (Mattie et al., 2010) demonstrating that Kinesin-2 through EB1 and APC moves plus ends of growing microtubules toward plus ends of existing microtubules, thus promoting uniform polarity of microtubules.The difference between this paper and the original paper and Mattie et al. is that here the authors are mostly looking at the growth of microtubules originated in the cell body and entering the axons while Mattie et al. studied uniform (minus-end-out) microtubule polarization in dendrites. Unfortunately, neither of these two papers addresses the key question of how the polarity in the dendrites and in the axons is first established, but only the question of how it is maintained. Whether the advances of this part of the manuscript justify its publication is not quite clear. What would make this part of the paper stronger is the imaging of EB1 and microtubules at the same time (the authors can label microtubules by tau-GFP or Jupiter-GFP) to show that EB1 on plus ends of growing microtubules indeed tracks pre-existing microtubules.

This is a good point and we had planned to perform these experiments using the UAS-Jupiter-mCherry line sent by Clemens Cabernard. Alas, this has not been possible due to University closure. We have therefore added the following statement to the Discussion section: “Our Kap3 RNAi data suggests that Kinesin-2 is required to guide growing microtubules along a polarised microtubule network within the soma towards the axon and away from dendrites, although future experiments with dual-colour imaging of microtubules and EB1-GFP comets will help support this conclusion”. We propose to carry out this experiment when our University opens and to include the results in the follow up paper.

In addition, the hypothesis that APC serves as a link between the two should be tested and discussed.

Before University closure, Melissa Rolls had kindly sent us an APC RNAi line that she had previously published (Mattie et al., 2010), but she had cautioned that APC phenotypes are weak, perhaps due to a maternal contribution of APC protein (Mattie et al., 2010). Nevertheless, we managed to generate five videos before University closure and found that 48.1% comets enter dendrites. While this is nearly twice than in controls (now measured at 27.7% after increasing the N number), the difference is not quite significant at the 5% cut-off due to the low numbers (P=0.0775). We have therefore decided not to include this data in the new submission, preferring to add to the N number once the University re-opens and publish the results in the follow-up paper. We hope the reviewer’s will appreciate, however, that the difference is likely genuine and that the lack of significance is simply due to the low N-number for APC RNAi neurons.

Reviewer #2:[…]Major points:1) The authors describe localization of γ-tubulin-GFP to puncta, diffuse spots, branch points and dendritic bubbles in Figures 1-5. The functional significance of these localizations is not clear and largely unexplored in the manuscript.

We agree with the reviewer that we have not demonstrated a functional role for the γ-tubulin accumulations within the dendrites. We reported these accumulations because we considered that, given the current controversy regarding the role of dendritic Golgi outposts and/or branchpoints as nucleation centres, it would be useful to the field if we reported in detail the endogenous localisation pattern of γ-tubulin within dendrites (something that has not been done before). We felt providing functional data for these dendritic γ-tubulin accumulations would go beyond the scope of this manuscript. Given the differential patterns observed within the dendrites of class I and class IV neurons, it will take a very careful and extensive analysis to understand and compare their roles. While Melissa Rolls has now published that γ-tubulin localises to endosomes in branchpoints, this is only within class I neurons and the vast majority of their γ-tubulin localisation data is from ectopically expressed γ-tubulin-GFP (only one image, using our own γ-tubulin-sfGFP line, is included in their supplementary data) (Weiner et al., 2020). In addition, our data also shows that γ-tubulin is not just restricted to branchpoints within dendrites but localises within dendritic stretches and dendritic bubbles in Class I neurons which has not been reported before. In fact, our quantification, which is the first of its kind, shows that most γ-tubulin accumulations are found outside of branchpoints in this neuron type. Our data is also the first to show clear endogenous γ-tubulin puncta associated with Golgi outposts in the proximal branchpoints of class IV neurons and makes it clear that this Golgi outpost associated γ-tubulin is restricted to these proximal branchpoints.

Collectively, robust analysis of endogenous γ-tubulin localisation in both class I and class IV neurons is of high value to the field. It will help explain the current controversy surrounding Golgi outposts and will provide important information for others to take into account in their future research. We therefore hope the reviewer agrees that it is worth keeping this data within the new submission.

We also point out that from Figures 3 onwards we are not just analysing these dendritic puncta, we are also analysing the somatic Golgi.

2) The first functional data is presented in Figure 6 with cold treatment of larvae to depolymerize microtubules. However, the key control of showing the treatment actually does depolymerize microtubules is missing.

We have summarised here several of the reviewers’ points that refer to their concern about whether our data proves that microtubules are nucleated from the somatic Golgi. We kindly refer the reviewer to the detailed discussion of this concern in the response to reviewer 1’s point 1 above.

3) The data in figure showing that Kinesin-2 may help prevent microtubules growing into dendrites is intriguing but based on quite low numbers of events and only one RNAi condition. This part of the manuscript is not very closely related to the rest of the work.

We had originally analysed 9 control and 6 Kap3 RNAi neurons. We therefore made and analysed four more control and three more Kap3 RNAi videos (now 13 control and 9 Kap3 RNAi videos in total) and this has increased the total number of comets approaching dendrites from 18 to 47 in controls and from 83 to 114 in Kap3 RNAi. Importantly, the percentage of comets that enter dendrites remains similar (a change from 22% to 28% in controls and from 59% to 57% in Kap3 RNAi) and the difference is still significantly different, supporting our model that plus end-associated Kinesin-2 is necessary for dendritic exclusion of outward growing microtubules. The new data is included in the new submission.

We have also now performed the experiment with a Klp64D RNAi line, which along with the Kap3 RNAi line has been validated previously (Mattie et al., 2010). From the ten Klp64D RNAi videos we have made there were 80 comets that approach dendrites and 56 entered (70.0%). We have therefore now reported this data in the new submission by including the following statement in the main text: “This affect was even more striking when knocking down a different Kinesin-2 component, Klp64D, where 56 of the 80 comets (across 10 videos) that approached a dendrite could enter (70.0%; p<0.001; Figure 7—video 3)”, as well as including an example video (new Figure 7—video 3).

In terms of whether this part of the manuscript is closely related to the rest of the work, it is of course closely related to Kinesin-2 mediated guidance of plus ends within soma (which has 2 facets: guidance towards the axon and exclusion of comets from dendrites). We believe the reviewer must therefore be referring to the work surrounding the localisation of γ-tubulin-GFP within the neurons, to which the reviewer refers to in comment 1 above. To help convince the reviewer that the work is related, we would like to explain the logical flow of our experiments: 1) We first analysed the localisation pattern of endogenous γ-tubulin-GFP within da neurons in an unbiased manner; 2) We found that γ-tubulin localised most strongly to the somatic Golgi rather than to sites within dendrites (although we document the differences in dendritic localisation between class I and class IV neurons); 3) We therefore tested whether the somatic Golgi nucleated microtubules by examining the growth of EB1-GFP comets within the soma – our data suggested that the Golgi did nucleate microtubules and that the microtubules were preferentially growing towards axons; 4) We therefore wondered what the fate of these microtubules may be, and found they tended to turn within the soma and grow into axons. In contrast, we also found that the microtubules that did happen to grow towards dendrites did not readily enter – we therefore wanted to know why; 5) We considered that Kinesin-2 plus end guidance might regulate both plus-end turning and exclusion from dendrites, as Kinesin-2 might engage with a polarised microtubule network within the soma to guide microtubules towards the axon, but might also engage with oppositely polarised microtubules at dendrite entry sites to generate a stalling force and depolymerisation of the outward growing microtubule; 6) We therefore depleted Kinesin-2 components, observing that Kinesin-2 was indeed necessary for both guiding microtubules towards axons and excluding them from dendrites.

Overall, we argue that this logic is sound and that the data is sufficiently related and of importance for the field of microtubule regulation in neurons to be published within the same paper.

Points on individual figures:Figure 1: The authors distinguish γ-tub-GFP puncta and diffuse patches. How were these distinguished? Is it known whether one is more functionally relevant than the other? In the example images in B, which are puncta, and which are diffuse? I did not see a description of how the different types of signals were quantitated although numbers are mentioned in the text.

When quantifying, we visually categorised them into one or other group depending whether the puncta were clear and obvious or not. If they were clear and obvious, then they were categorised as puncta. If puncta and diffuse patches did co-exist, we defined them as puncta. We have now included a description of how we quantified the puncta and diffuse patches in the Materials and methods. In Figure 1B, all are diffuse except for the branchpoint on the far left of the panel. We do not know whether puncta or diffuse patches are more functionally relevant. It could be that both types localise to endosomes (as reported in Weiner et al., 2020) and the pattern of γ-tubulin depends simply on whether the endosomes are clumped together or more evenly spread. Some puncta may also localise to Golgi outposts, as we show in Figure 1D, but our data would suggest this is rare in class I.

In the legend it is noted that images in B and C are from living animals. It is a bit confusing whether A is also live. It looks like the signal in fixed neurons is more punctate than in live ones. Is this an accurate impression? If so, it would be good to mention in the text.

Figure 1A is also from a live specimen, and this was mentioned in the legend, but we now also mention it in the main text: “The most striking and obvious localisation of γ-tubulin-GFP within both class I and class IV neurons was as multiple large bright puncta within their soma (see images of living neurons in Figure 1A and Figure 2A, and of fixed and stained neurons in Figure 3A-C)”. The weak dendritic γ-tubulin-GFP signal can easily be missed, as it is close to background levels, and it is likely harder to see in fixed samples, potentially giving the impression that the signal in fixed neurons is more punctate. We do not see any obvious increase in the number of puncta in fixed images compared to live images. As we do not see a consistent difference, we have not mentioned anything on this point in the new submission.

Figure 2: It looks like there is a movement artifact at the right side of the images in A and A' (double spots), so it might be good to choose a different example.

The reviewer is correct, and we thank the reviewer for pointing it out. This movement is due to muscle contractions and is not unusual when imaging living animals. It tends to occur close to the segment boundaries. As the shift in this particular image is confined to the extreme right hand-side of the image, affecting only a small part of the neuron, we have cropped out the small region that had shifted.

The authors conclude: "Collectively, our data show that Golgi outposts within class IV neuron branchpoints close to the soma frequently associate with γ-TuRCs," However, no data on γ-TuRCs was presented – are the authors assuming that the spots where they see γ-tubulin-GFP are γ-TuRCs? This claim should be supported either by looking at other γTuRC subunits or with functional analysis of nucleation.

γ-tubulin is the most commonly used marker for γ-TuRCs, at least in part because of its high stoichiometry within the complex. γ-tubulin will normally represent γ-TuRCs – we do not know of any reports of γ-tubulin accumulations within cells that do not represent γ-TuRCs. But of course, we cannot rule out this possibility. We have recently managed to generate Grip91-sfGFP endogenously-tagged lines, but unfortunately we did not have time to fix, stain and image them prior to University closure. Nevertheless, while Grip91 is the protein with the next highest stoichiometry in the complex, it is not actually γ-TuRC specific as it is also present in γ-TuSCs (the smaller subcomplex of γ-TuRCs). To be sure of γ-TuRCs, it is therefore best to use a γ-TuRC-specific component, such as Grip75, Grip128, or Grip163, but these have a much lower stoichiometry and so the signal at the Golgi will be much weaker. Our initial observations of Grip75-GFP in living neurons that we were able to make before University closure suggested that the signal is very weak even at the somatic Golgi. Thus, rather than directly addressing this point experimentally, we have toned down the statement about γ-TuRCs localising to Golgi outposts, as well as only referring to the accumulation as “γ-tubulin” rather than “γ-TuRCs” throughout the preceding paragraph. The statement at the end of the paragraph now reads: “Collectively, our data show that Golgi outposts within class IV neuron branchpoints close to the soma frequently associate with γ-tubulin, but that the majority of Golgi outposts do not. Whether this proximal Golgi outpost associated γ-tubulin represents fully functional γ-TuRCs remains to be tested.”

While quantitation of Golgi and γ-tub-GFP is mentioned in the text, it is not clear how the quantitation was performed, and I could not find a description of it in the Materials and methods. It would be helpful to include graphs of the quantitation.

We apologise for not including this in the original submission. We have now included the details in the Materials and methods of the new submission. We thought that adding graphs would make the figures more crowded than they needed to be and so have not included them. We are willing to do so, however, should the reviewer deem it absolutely necessary.

Figure 3. The authors stain γ-tubulin-GFP larvae with HRP and antibodies to GM130 and observed some intriguing patterns in the cell body. They conclude "this suggests that γ-tubulin-GFP localises on the rims of the cis-Golgi stack." It would be helpful to support this conclusion with other markers. Is the association with cis-Golgi higher than that with a trans-Golgi marker? Without labeling other parts of the Golgi, or surrounding structures like ER exit sites, it is hard to know whether the spots around the Golgi are most closely associated with cis-Golgi. It is particularly important to pin this down as Plp, which was previously shown to link nucleation sites to the Golgi, seems not to have a role here (Figure 5).

We agree with the reviewer and have stained γ-tubulin-GFP larvae with HRP and antibodies against GM130 (cis-Golgi) and Arl1 (trans-Golgi) and obtained four channel images. The new data clearly shows that Arl1 is offset from both γ-tubulin-GFP and GM130, which overlap (new Figure 3D). While we used the HRP signal to identify neurons during analysis, we have not included this channel in the new figure for simplicity. Now that we are more certain that γ-tubulin localises to the cis-Golgi, we have also slightly modified the cartoon images of Golgi stacks in our model figure, now Figure 8C.

We make the point again here that now we have proof of the asymmetric localisation of γ-tubulin to the cis-Golgi, we can be more certain that the somatic Golgi nucleates microtubules, as the asymmetric localisation fits well with the observed asymmetric microtubule growth from the Golgi stacks, which is known to also occur in mammalian cells.

Figure 6. The authors examine the site of new EB1 comet formation in the cell body of class I neurons. How do they know which comets are the result of catastrophe rescue and which are nucleation? If they want to link comet formation to nucleation, it is important to show that it is altered in γ-tubulin mutants.

We agree that this would have been a nice experiment and we had planned to deplete γ-TuRCs and examine the effect on EB1-GFP comets. Alas, the University closed while the fly lines for this experiment were still being generated. It is important to note, however, that this experiment may not technically be possible. Our past experience has shown that effectively depleting γ-TuRCs within larval da neurons can be challenging. We believe this is due to a maternal contribution of γ-tubulin-37C, because we know that γ-tubulin-23C null mutant larvae can live up until the end of the third instar larval stage (unpublished observations). The persistence of maternal γ-tubulin-37C is consistent with results from other groups where attempts to reduce γ-tubulin-23C in class I da neurons have not produced dendritic morphology defects (Nguyen et al., 2014; Weiner et al., 2020), even though γ-tubulin has now been proposed to function at branchpoints in class I neurons (Weiner et al., 2020). Nevertheless, we plan to perform this experiment after the University re-opens and include the results in a future follow-up paper.

Some comets were determined to originate from the Golgi. The marker used for the Golgi was ManII, which earlier was described as not localizing cleanly to the Golgi in Class I neurons, which were used for this assay. Overall it is very difficult to know whether the new comets counted in this figure represent nucleation and whether they derive from the Golgi.

While UAS-ManII-GFP is not a reliable marker of Golgi outposts within class I dendrites, it always colocalises with HRP within the soma. We therefore now show an example of this in the new Figure 1—figure supplement 1D. Thus, UAS-ManII does mark genuine Golgi structures in the soma, validating its use in the comet analysis.

A bias towards the axon would be very easy to explain based on catastrophe rescue of microtubules that derive from the dendrites, as these grow into the soma towards the axon.

We thank the reviewer for pointing this out and it is a caveat we have highlighted in the new manuscript. Nevertheless, we make our view clear that we believe the effect is minimised by analysing only those comets that originate from the Golgi: “While it is possible that the direction of comet growth can be influenced by comets growing into the soma from the dendrites, which will tend to grow towards the axon, we minimised this effect by quantifying only those comets that originate from Golgi stacks. The comets we analysed are therefore more likely to represent true growth events from the Golgi rather than catastrophe and regrowth of microtubules that originated from dendrites.”

For the microtubule cold depolymerization it is important to show that stable microtubules are actually depolymerized in neurons. They tend to be very refractory to cold or drug-induced depolymerization and unless stable microtubules really are eliminated by this treatment regrowth could be from remaining microtubules rather than nucleation. An additional confirmation that this assay is a good readout of nucleation would be to perform it in γ-tubulin mutant or knockdown neurons. Their statement "any comets that appear a significant amount of time after warming represent new nucleation events" requires significantly more validation.

We agree with the reviewer that we had not made this point clearer in the original paper, and this comment made us realise that we needed to explain ourselves better, especially as we feel this observation is one of the strongest p. We have therefore removed the original sentence and included instead a detailed explanation within the Results section: “In our opinion, the best evidence for microtubule nucleation occurring at the somatic Golgi comes from the observation that several comets emerged from Golgi stacks relatively late after warming. […] This strongly suggests that these late emerging comets represent genuine nucleation events from the Golgi, although it is impossible to rule out that they could have been generated by regrowing microtubules that were originally out of focus.”

This section of the text also lacks references, for example on whether cold treatment has previously been shown to depolymerize neuronal microtubules.

To our knowledge, this has not been tried in *Drosophila* neurons in vivo, although cold-induced depolymerisation has been carried out in other cell types. Of note, cold-stable microtubules exist within mammalian neurons, but this relies on MAP6, which is restricted to vertebrates (see Delphin et al., 2012, JBC). We have therefore now included the following statement and citations in the Results section: “Cooling-warming microtubule nucleation assays have previously been performed in various systems, including *Drosophila* embryos (Hayward et al., 2014), *Drosophila* S2 cells (Bucciarelli et al., 2009), and in mammalian cells (Torosantucci et al., 2008). […] We therefore presumed that cooling would result in the depolymerisation of dynamic microtubules within *Drosophila* neurons, allowing us to correlate the position of new comet growth with nucleation sites.”

Figure 7. The title of this section is "Kinesin-2-dependent plus-end turning and dendritic exclusion maintain microtubule polarity in axons and dendrites." Kinesin-2 also functions more distally in dendrites to control direction of microtubule growth. The section title attributes changes in polarity to the cell body rather than this function. Can the data the authors present link dendrite polarity changes to a function in the cell body rather than in the dendrites? Is it because they only analyze the proximal dendrite?

Yes, this is because we only analyse the proximal dendrite. This was mentioned in the figure legend, but we now realise that we need to make this clear in the main text, and that we also need to change the section title to more adequately reflect our findings. We have changed the title to: “Growing microtubules within the soma are guided towards the axon while being excluded from entering dendrites in a Kinesin-2-dependent manner”. We have also changed the final concluding statement to read: “We conclude that Kinesin-2 is required to guide growing microtubule plus ends within the soma towards the axon and to prevent them from entering dendrites, which is essential in order to maintain minus-end-out microtubule polarity within proximal dendrites.”

There is no data on microtubule polarity in axons, although the section title states that Kinesin-2 has a role in maintaining axon microtubule polarity. Are there changes in axonal polarity in the Kap3 knockdown?

We apologise for not making ourselves clearer. We believe there is a role for promoting plus-end out polarity in axons because Kinesin-2 guides the growing microtubules towards axons, which should in theory lead to more microtubules growing into the axon and contributing to the plus-end-out pool. We agree, however, that there is no obvious effect on axon polarity in Kap3 RNAi and have therefore removed the reference to the axon in the section title. The title is now: “Growing microtubules within the soma are guided towards the axon while being excluded from entering dendrites in a Kinesin-2-dependent manner”.

The major finding here, that comets normally do not grow easily in dendrites is interesting. It would be helpful to have stronger data to link Kap3 steering to keeping microtubules away from the dendrite; the increase in dendrite approaches is not quite significant (c). It would be very helpful to analyze more animals to see whether this will hold up, and the numbers of plus ends analyzed is quite low.

We agree and have now analysed more control (13) and Kap3 RNAi (9) neurons. With the increased number of comets available to analyse, the difference between the proportion of comets approaching dendrites in control (7%) versus Kap3 RNAi (11%) is now significant (p=0.0098). We can therefore conclude that Kinesin-2 is required to prevent growing microtubules from entering dendrites but is also required to steer growing microtubules away from dendrites in the first place. We now refer to this result within the Results and Discussion.

The increase in successful entries into the dendrite (D) in the Kap3 RNAi is more convincing, but how does this relate to the model?

This actually agrees perfectly with the model and is, to us, the most important result of the paper. This comment made us realise that we need to explain our model much more clearly in the new submission. We have therefore placed the model in a new Figure 8, that also includes a new section (Figure 8A, B) to help explain how Kinesin-2 differentially regulates plus end growth in different contexts, and we have also included an expanded discussion of the model in the Discussion in the paragraph starting: “A key feature of our model is the action of plus-end associated Kinesin-2”

Similarly, if Kinesin-2 was helping microtubules grow along parallel microtubules, would it not be expected that the successful entries into the axon in D would go down?

While Kinesin-2 may guide microtubules along parallel microtubules, there is no requirement for Kinesin-2 in microtubule growth per se. Rather, Kinesin-2 is required for microtubules to engage with and turn along the path of the other microtubules (i.e. when guided towards the axon entry site), and also to prevent microtubules growing parallel to oppositely polarised microtubules (i.e. preventing them from entering dendrites). Once a microtubule has reached the axon entry site, its growth into the axon should in theory not require Kinesin-2, which fits with our data. With our new increased N number, we find that the moderate increase in the number of comets that enter the axon in Kap3 RNAi neurons is actually statistically significant. While this increase is not as dramatic as the increase in dendrite entry, we feel it also deserves an explanation. We have therefore included the following statement in the new submission: “Comets that did arrive at the axon entry site could still readily enter the axon after Kap3 RNAi (Figure 7D); there was actually a ~1.3-fold increase compared to controls in the proportion of comets entering the axon (72.2% versus 53.5%, p=0.007). […] Importantly, increased entry of growing microtubules into the axon has no major effect on microtubule polarity, as axons normally contain predominantly plus-end-out microtubules.”

In the text the authors mention some additional data on turning within the cell body that is not shown in the figure. It would be good to have this in the figure as it adds additional support to the hypothesis that Kinesin-2 is functioning to control microtubule growth in the cell body.

I think the reviewer may have missed this data – it is included as a graph in Figure 7E.

[Editors’ note: what follows is the authors’ response to the second round of review.]

Reviewer #1:The authors have done everything they could under the current conditions. In normal times, I would certainly require simultaneous imaging of plus ends and a microtubule marker at the same time, this would make the paper much stronger. However, this type of imaging is not trivial, and it would be unfair to require it now. I think we can accept the manuscript.

This reviewer felt that the paper could now be accepted but would have liked to see simultaneous imaging of plus ends and a microtubule marker. We refer to the importance of this experiment in the Discussion: “A key feature of our model is the action of plus-end-associated Kinesin-2. Our Kap3 RNAi data suggests that Kinesin-2 is required to guide growing microtubules along a polarised microtubule network within the soma towards the axon and away from dendrites, although future experiments with dual-colour imaging of microtubules and EB1-GFP comets will help support this conclusion.” We plan to do this experiment in the near future, by imaging EB1-GFP in a Jupiter-mCherry background and include the results in the proposed *eLife* advance paper.

Reviewer #2:This manuscript is a resubmission of a previous version that was rejected. Unfortunately, the key functional experiments that were suggested in the previous comments were not performed because of Covid-related lab closure. While some leeway seems reasonable for manuscripts close to acceptance, asking key data to be overlooked for a rejected manuscript seems a stretch. While the authors were able to increase the n on one experiment and provide some additional staining, as they mention in the rebuttal many of the important experiments have not been done yet. Therefore, the limitations in the original submission remain.

When this reviewer wrote their independent review, they still had certain reservations about the lack of functional data for our observations of γ-tubulin localisation within dendrites and our conclusion that microtubule nucleation occurs at the somatic Golgi. These concerns are similar to their original concerns that we were unable to fully address due to the 3-month lockdown period. As requested, below we address these concerns textually as best possible and highlight the textual changes we have made in the manuscript (several of which were already present in the re-submitted paper). We also highlight the experiments that we plan to include in the aforementioned *eLife* advance paper.

Examples of key issues that have not been changed in the new version:Much of the data remains descriptive without functional backup. For example, γ-tubulin localization to puncta vs diffuse concentrations is described, but it is not clear whether one type of localization is functional, or both are. Similarly, γ-tubulin is described as localizing to dendrite bubbles, and again this is unconnected to function.

Our data on γ-tubulin-GFP localisation within dendritic arbors is the first to fully describe and carefully quantify the localisation pattern of endogenously-expressed γ-tubulin within class I and class IV da neurons. We did not provide any functional data for these observations because this would go beyond the scope of the paper, which focusses on microtubule regulation within the soma of these neurons. Nevertheless, we strongly believe that quantifying and comparing the localisation of endogenously-tagged γ-tubulin-GFP within the dendrites of class I and class IV da neurons is important and should be reported to the community, particularly due to the ongoing debate over whether Golgi outposts act as MTOCs or not. Given the reviewer’s continued concern, we now specifically mention in the Results section that we lack functional data for these observations: “Nevertheless, this dendritic localisation that we observe likely represents the recently reported recruitment of γ-tubulin to endosomes at branchpoints (Weiner et al., 2020)” to “It is possible that the localisation of endogenous γ-tubulin-GFP at branchpoints represents the recently reported recruitment of γ-tubulin to endosomes at branchpoints that provide a platform for microtubule nucleation (Weiner et al., 2020). We have not tested here, however, whether the diffuse or punctate γ-tubulin-GFP signals, or both, are functionally important at branchpoints.” and we have added “It remains to be tested whether the accumulation of γ-tubulin within these bubbles is functionally relevant.”

Analysis of microtubule behavior in the cell body of γ-tubulin mutants is important to demonstrate that the comets initiating near Golgi are due to nucleation. In their live experiments with Golgi and EB1 (Figure 6) about half of the cell body looks like it is occupied by the Golgi marker. I did not see any quantitation to show that new comet formation is enriched on Golgi compared to other areas of cytoplasm. It is important to do this in control and γ-tubulin mutant animals.

This comment relates to the reviewer’s continued concern as to whether the observed asymmetric localisation of γ-tubulin at the somatic Golgi, asymmetric microtubule growth from the somatic Golgi, and emergence of microtubule growth after cooling-warming from the somatic Golgi, is really representative of microtubule nucleation from the somatic Golgi. While the Editors’ view is that “we recognize that this is generally a very difficult question to address (localization-specific function of a given protein). And other circumstantial evidence provided in the paper is sufficiently strong to propose the function of Golgi-localized microtubule nucleation” we would still like to respond to the reviewer’s comment and highlight the textual changes we have made to the manuscript.

We agree with the reviewer that it will be important in future to try to deplete γ-TuRCs (something that is not trivial, as discussed in our last rebuttal) and then assess whether there is a reduction in comets emerging from the Golgi. We plan to perform this experiment in the near future, once the fly stocks are ready, and include the results in the proposed *eLife* advance paper. We think that comparing the frequency of comets emerging from the Golgi versus those emerging within the cytosol in wild-type neurons will not necessarily help determine whether microtubules are nucleated from the Golgi for at least two reasons. Firstly, Augmin-mediated nucleation could be occurring from the side of microtubules within the cytosol in addition to nucleation from the Golgi. If the rate of nucleation via Augmin was relatively high, this might make it appear that comets emerging from the Golgi were less significant, even if they were also being nucleated from the Golgi by γ-TuRCs. We mention the possibility of Augmin-mediated nucleation. Secondly, the results will be affected by plus-end dynamics post-nucleation i.e. the rate of catastrophe-rescue events where a microtubule partially depolymerises and then re-grows. We observe comets with relatively short tracks within the soma, indicating that the microtubules can be highly dynamic. This could lead to a relatively high frequency of new comets emerging within the cytosol (due to regrowth events), even if all microtubules were nucleated only from the Golgi. As mentioned in our last rebuttal, the fact that comets emerge asymmetrically from the Golgi (coupled with the asymmetric localisation of γ-tubulin at the Golgi), and that multiple comets emerging from the same Golgi stack emerge in a similar direction, lends strong support to the idea that microtubules are indeed nucleated from the Golgi.

We had already made several textual changes in our re-submitted paper that show we are being cautious about our conclusion that microtubules are nucleated from the somatic Golgi:

– In the title and Abstract we use the term “microtubules originate” rather than saying “microtubules are nucleated” when referring to the somatic Golgi.

– Results: “While not definitive, the similarity in the direction of comets emerging repeatedly from the same Golgi stack is indicative of consecutive microtubule nucleation events.”

– Discussion: “the model states that microtubules are nucleated asymmetrically from the somatic Golgi”. The important point being that this is a model and not a statement of fact.

– Discussion: “After showing that γ-tubulin-GFP localised to the somatic Golgi, we tracked EB1-GFP comets from the Golgi and found that they grew with an initial directional preference towards the axon, suggestive of asymmetric nucleation”.

– Discussion: “It is possible that a fraction of comets that originate from the somatic Golgi represent microtubules that were re-growing after partial depolymerisation, where the re-growth event happened to overlap a Golgi stack. However, several pieces of evidence suggest that the majority of the Golgi-derived comets represent nucleation events…”

– Discussion: “In summary, when we consider all of our data as a whole, we believe the most parsimonious conclusion is that microtubules are nucleated asymmetrically from the somatic Golgi.”

The temperature shift assay remains difficult to interpret without additional information about what microtubules remain after cooling. There also does not seem to be quantitation in the figures to support conclusions made from this data.

We agree that it will be important to test the extent of microtubule depolymerisation during cooling, and we plan to do this experiment in the near future, reporting the results in the *eLife* advance paper. Nevertheless, interpreting what happens to microtubules during cooling may be problematic due to a likely mix of stable (cold-insensitive) and dynamic (cold-sensitive) microtubule populations. We explained this in our last rebuttal and mention it in the manuscript. While there is no quantification in the figures (it is difficult to get sufficient numbers due to the thermal fluctuations leading to loss of the focal plane during temperature transitions, which is mentioned in the Discussion, we now report the following numbers in the text: “Within 20s of warming in Figure 6—video 2, 4 comets emerged from Golgi structures (green-labelled comets, Figure 6E; Figure 6—video 2), while 2 comets emerged away from the Golgi (purple-labelled comets, Figure 6E; Figure 6—video 2). During the 2 minutes and 12 seconds between warming and the end of the video, a total of 8 comets emerged from the Golgi with 5 emerging from non-Golgi sites.” We go on to postulate that the non-Golgi associated comets that emerge initially after warming could be due to Augmin mediated nucleation, Golgi-nucleated comets that emerge into the focal plane post nucleation, or re-growing microtubules that had not fully depolymerised. This text remains unchanged from the last version.

We also want to point out the textual changes we had already made to the re-submitted paper that show we are being cautious about our interpretation of the cooling-warming assay:

– Results: “When the appropriate focal plane was reached shortly after warming, we could observe EB1-GFP comets emerging from the ManII-mCherry-labelled Golgi structures (green-labelled comets, Figure 6E; Figure 6—video 2), suggestive of Golgi-located microtubule nucleation events.”

– Results: “This strongly suggests that these late emerging comets represent genuine nucleation events from the Golgi, although it is impossible to rule out that they could have been generated by re-growing microtubules that were originally out of focus.”